# Comprehensive Analysis of Granzymes and Perforin Family Genes in Multiple Cancers

**DOI:** 10.3390/biomedicines13020408

**Published:** 2025-02-07

**Authors:** Manvita Mareboina, Katrina Bakhl, Stephanie Agioti, Nelson S. Yee, Ilias Georgakopoulos-Soares, Apostolos Zaravinos

**Affiliations:** 1Institute for Personalized Medicine, Department of Biochemistry and Molecular Biology, The Pennsylvania State University College of Medicine, Hershey, PA 17033, USA; mmareboina@pennstatehealth.psu.edu (M.M.); kbakhl@pennstatehealth.psu.edu (K.B.); 2Department of Life Sciences, School of Sciences, European University Cyprus, Nicosia 1516, Cyprus; sa231751@students.euc.ac.cy; 3Cancer Genetics, Genomics and Systems Biology Laboratory, Basic and Translational Cancer Research Center (BTCRC), Nicosia 1516, Cyprus; 4Department of Medicine, Division of Hematology-Oncology, Penn State Health Milton S. Hershey Medical Center, Next-Generation Therapies Program, Penn State Cancer Institute, Hershey, PA 17033, USA; nyee@pennstatehealth.psu.edu

**Keywords:** granzyme a, granzyme b, granzyme k, perforin 1, pan-cancer, immune infiltration, immunotherapy, tumor expression levels, immunotherapy, drug sensitivity

## Abstract

**Background/Objectives:** Cancer remains a significant global health concern, with immunotherapies emerging as promising treatments. This study explored the role of perforin-1 (*PRF1*) and granzymes A, B and K (*GZMA*, *GZMB* and *GZMK*) in cancer biology, focusing on their impact on tumor cell death and immune response modulation. **Methods:** Through a comprehensive genomic analysis across various cancer types, we explored the differential expression, mutation profiles and methylation patterns of these genes, providing insights into their potential as therapeutic targets. Furthermore, we investigated their association with immune cell infiltration and pathway activation within the tumor microenvironment in each tumor type. **Results:** Our findings revealed distinct expression patterns and prognostic implications for *PRF1*, *GZMA*, *GZMB* and *GZMK* across different cancers, highlighting their multifaceted roles in tumor immunity. We found increased immune infiltration across all tumor types and significant correlations between the genes of interest and cytotoxic T cells, as well as the most significant survival outcomes in breast cancer. We also show that granzymes and perforin-1 are significantly associated with indicators of immunosuppression and T cell dysfunction within patient cohorts. In skin melanoma, glioblastoma, kidney and bladder cancers, we found significant correlations between the genes of interest and patient survival after receiving immune-checkpoint inhibition therapy. Additionally, we identified potential associations between the mRNA expression levels of these genes and drug sensitivity. **Conclusions:** Overall, this study enhances our understanding of the molecular mechanisms underlying tumor immunity and provides valuable insights into the potential therapeutic implications of *PRF1*, *GZMA*, *GZMB* and *GZMK* in cancer treatment.

## 1. Introduction

Cancer remains a pervasive global health challenge and a primary cause of death [1], with its intricate biology posing significant therapeutic obstacles. In recent years, immunotherapy has emerged as a promising weapon for cancer treatment, harnessing the power of the immune system to combat malignancies [2]. Somatic mutations shape the development of cancer and influence its initiation, progression, and response to therapies [3,4,5,6,7]. Perforins and granzymes have gained increasing interest due to their essential role in producing an immune response against cancer cells [8,9,10]. They are both stored in cytotoxic lymphocytes, assisting in the cell-mediated immune response against cancer cells upon their release [11]. Perforin, a glycoprotein typically stored in natural killer (NK) cells and cytotoxic T cells (CTLs), facilitates pore formation in the target cells, promoting apoptosis and facilitating the entry of granzymes [11,12]. Granzymes, which are located in cytolytic granules, are proteases also typically found in NK cells and CTLs [12,13,14]. Granzymes and perforins are released through granule exocytosis; subsequently, perforin creates pores within the target cell, after which, granzymes enter the cell’s cytoplasm [11,12]. Following entry, an enzymatic cascade is initiated by granzymes, and which leads to apoptosis [11,12]. It has been shown that granzyme A promotes IL-1β activation and cytokine release, while granzyme B contributes to detachment-induced apoptosis by cleaving components of the extracellular matrix [11].

The genomics behind the role of these three immune factors has also been analyzed in cancer, inflammation and immune surveillance. Specifically, granzyme A (GZMA), granzyme B (GZMB), granzyme K (GZMK) and perforin-1 (PRF1) [8,9,10,15] do not only induce apoptosis and modulate immune response within the tumor microenvironment (TME), but also have other diverse roles, such as potentially regulating homeostasis post-infection [11,16]. Research has shown that granzymes can degrade microbial components needed for proliferation, thus showing their antimicrobial effects. This phenomenon has been particularly observed in virus-infected cells, unicellular parasites and intracellular bacteria, with reactive oxygen species (ROS) and mitochondria being key players in the cascade [15]. Moreover, perforin-1 and granzymes have been used as a potential therapeutic target in diabetes mellitus type I given their role in destroying beta cells and rejecting allogeneic islet grafts [17].

In cancer and immune surveillance, perforin and granzymes play crucial roles in mediating cancer cell death [11]. Although there is no known connection between granzymes and cancer risk, the absence of perforin raises the chances of tumor development, supporting the notion of a degree of redundancy in granzyme function in tumor destruction. In a few experiments, not only were perforin-deficient mice unable to assist the immune system in fighting viruses, but their immune cells were also unable to eliminate tumor cells in vitro [11]. On the other hand, limited research has shown that tumor cells may use the perforin and granzyme cascade mechanism to advance tumor proliferation [11]. Thus, an extensive analysis of *GZMA*, *GZMB*, *GZMK* and *PRF1* may not only provide a greater understanding of their role in cancer biology, but also reveal their similarities and differences across different cancer types. Here, we examined the role of *GZMA*, *GZMB*, *GZMK* and *PRF1* in the immune response, immune infiltration, patient survival, drug sensitivity and therapeutic responses in multiple cancers.

## 2. Materials and Methods

### 2.1. Differential Expression

#### 2.1.1. *GZMA*, *GZMB*, *GZMK* and *PRF1* Expression in Different Cancers

To investigate the differential expression of *GZMA*, *GZMB*, *GZMK* and *PRF1* across 14 cancer types and their paired normal tissues, we utilized the Gene Set Cancer Analysis (GSCA) platform (https://guolab.wchscu.cn/GSCA//#/) (accessed on 10 May 2024) [18]. Data normalization and batch correction were performed using RSEM (RNA-Seq by Expectation Maximization) for mRNA expression, ensuring comparability across samples and batches. Log_2_ transformation with TPM normalization was employed for the expression analyses, primarily utilizing data sourced from the GEPIA2 platform (https://gepia2.cancer-pku.cn/) (accessed on 20 May 2024), which integrates TCGA and GTEx datasets. While the GSCA database includes data from 33 cancers, expression data for all four genes were available for 14 cancers. Consequently, the differential expression analysis (Figure 1a) reflects these 14 cancer types due to limitations in the available data for the paired differential expression boxplots. However, in the rest of the study, the data from all 33 cancers were utilized when available, ensuring comprehensive analyses wherever possible. A complete list of the databases utilized and their point of access can be found in Table 1. The different cancer types and the corresponding sample numbers for all 33 cancer types in the database were as follows: Adrenocortical Carcinoma (ACC), 79; Bladder Urothelial Carcinoma (BLCA), 427; Breast Invasive Carcinoma (BRCA); 1218; Cervical Squamous Cell Carcinoma and Endocervical Adenocarcinoma (CESC); 310; Cholangiocarcinoma (CHOL), 45; Colon Adenocarcinoma (COAD), 329; Lymphoid Neoplasm Diffuse Large B-Cell Lymphoma (DLBC), 48; Esophageal Carcinoma (ESCA), 196; Glioblastoma Multiforme (GBM), 174; Head and Neck Squamous Cell Carcinoma (HNSC), 566; Kidney Chromophobe (KICH), 91; Kidney Renal Clear Cell Carcinoma (KIRC), 606; Kidney Renal Papillary Cell Carcinoma (KIRP), 323; Acute Myeloid Leukemia (LAML), 173; Brain Lower-Grade Glioma (LGG), 534; Liver Hepatocellular Carcinoma (LIHC), 424; Lung Adenocarcinoma (LUAD), 576; Lung Squamous Cell Carcinoma (LUSC), 554; Mesothelioma (MESO), 87; Ovarian Serous Cystadenocarcinoma (OV), 309; Pancreatic Adenocarcinoma (PAAD), 183; Pheochromocytoma and Paraganglioma (PCPG), 187; Prostate Adenocarcinoma (PRAD), 550; Rectum Adenocarcinoma (READ), 105; Sarcoma (SARC), 265; Skin Cutaneous Melanoma (SKCM), 474; Stomach Adenocarcinoma (STAD), 450; Testicular Germ Cell Tumors (TGCTs), 156; Thyroid Carcinoma (THCA), 572; Thymoma (THYM), 122; Uterine Corpus Endometrial Carcinoma (UCEC), 201; Uterine Carcinosarcoma (UCS), 57; and Uveal Melanoma (UVM), 80. To quantify the degree of differential expression, the fold change was computed as the ratio of the mean expression in the tumor tissues to that in the normal tissues. The statistical significance of these differences was assessed using the *t*-test, with further adjustments for false discovery rates (FDRs). FDR values ≤ 0.05 were considered statistically significant.

#### 2.1.2. Gene Expression and Survival

We divided the tumor samples into high and low expression groups using the median mRNA value. Subsequently, we utilized the R package *survival* to model the survival time and status within these two groups and create Kaplan–Meier plots. For each gene in every cancer, we applied the Cox Proportional-Hazards model and Logrank tests. Various prognostic indicators, such as overall survival (OS), progression-free survival (PFS), disease-specific survival (DSS) and disease-free interval (DFI) were assessed to distinguish survival disparities between the two expression groups.

#### 2.1.3. T Cell Signatures

Additionally, we assessed the connection between the expression of the genes of interest and the exhausted T cell signature [19] using the GEPIA2 platform (https://gepia2.cancer-pku.cn/) (accessed on 20 May 2024). We set a log_2_ fold change (Log_2_FC) cutoff of 1 and a *p*-value cutoff of 0.01 to identify statistically significant results. We used log_2_(TPM + 1) for the expression analysis. We further correlated cytolytic activity (CYT, as defined by Rooney et al.) [20] with resting and effector regulatory T cell signatures across different cancers. In this study, we initially analyzed T cell/NK cell activity across all the cancer types with data available on the GEPIA2 platform. However, the subsequent analysis focused on a subset of 11 cancers that demonstrated notable gene expression results. This selection allowed for a more detailed comparison of the four genes of interest across these cancer types, particularly in relation to their roles in the tumor microenvironment. The predominance of specific cancer types, such as breast carcinoma, in the dataset reflects the biological relevance of these cancers to the study objectives.

### 2.2. Immune Infiltration

#### 2.2.1. Pathway Expression

We distinguished between pathway activation and inhibition groups based on the median pathway activity scores (PASs). The pathway activity scores (PASs) were normalized using standard deviation scaling. For this analysis, Reverse Phase Protein Array (RPPA) data were sourced from the TCPA (https://www.tcpaportal.org/tcpa/) database [21] (accessed on 5 June 2024). This dataset encompassed information on the pathway activity scores for ten cancer-related pathways across 7876 samples derived from 33 cancer types within the TCGA database. The set of pathways included the following: the tuberous sclerosis complex/mammalian target of rapamycin (TSC/mTOR), receptor tyrosine kinase (RTK), RAS/mitogen activated protein kinase (MAPK), phosphatidylinositol 3-kinase (PI3K)/protein kinase B (AKT), Hormone ER, Hormone AR, EMT, DNA Damage Response, Cell Cycle and Apoptosis pathways.

To compute the PASs, the RBN RPPA data underwent a two-step process involving median centering and normalization using the standard deviation across all samples for each constituent. The PAS was determined as the sum of the relative protein levels of the positively regulating components minus those of the negatively regulating components within a specific pathway [22]. The samples were dichotomized into “high” and “low” groups based on the median gene expression. Differences in PASs between these groups were statistically evaluated using Student’s *t*-test, with *p*-values adjusted for the FDR. Significance was attributed to results with an FDR ≤ 0.05. In cases where the PAS for each gene in the “high expression” group exceeded that in the “low expression” group, it indicated an activating influence on a pathway, while the opposite scenario suggested an inhibitory effect [23]. The pathway activities for apoptotic pathways with low and high expression of the genes *GZMA*, *GZMB*, *GZMK* and *PRF1* were individually plotted for BRCA.

#### 2.2.2. Immune Infiltration

We assessed the association between gene mRNA expression and the presence of immune cell infiltrates within tumor tissues using Spearman correlation. The abundance of 24 distinct immune cell types was evaluated utilizing ImmuCellAI [24,25]. Estimates of the abundance of these 24 immune cell types relied on gene set signatures. Correlations were also plotted for the ten highest correlation values between the cancers and immune cell infiltrates for *GZMA*, *GZMB*, *GZMK* and *PRF1*.

#### 2.2.3. ESTIMATE

We predicted the tumor purity and the presence of infiltrating stromal/immune cells in the tumor tissues using ESTIMATE (Estimation of STromal and Immune cells in MAlignant Tumor tissues using Expression data). The ESTIMATE algorithm is based on single-sample Gene Set Enrichment Analysis (ssGSEA) and generates three scores: (1) a stromal score (that captures the presence of stroma in the tumor tissue); (2) an immune score (that represents the infiltration of immune cells into the tumor tissue); and (3) an estimate score (that infers tumor purity) [26]. All of the estimate score ranges are included in Appendix A.

### 2.3. Mutation Analysis

#### 2.3.1. SNV Mutations

We retrieved single-nucleotide variant (SNV) data from a total of 10,234 samples encompassing 33 distinct cancer types. Our analysis focused on seven mutation types, categorized as potentially deleterious mutations: missense, nonsense mutations, frameshift insertions/deletions, splice site mutations, and in-frame insertions/deletions. Exclusively deleterious mutations were categorized as mutants in the modules pertaining to the SNV data. An oncoplot was also generated for SNV mutations using the R studio Maftools package, which shows a comprehensive view of the top 10 mutated genes from the provided gene set within the selected cancer samples. Samples with at least one SNV were included in the oncoplot.

#### 2.3.2. CNV Mutations

Copy number variation (CNV) data, comprising 11,495 samples, were analyzed using GISTIC2.0, spotlighting significant regions of amplification or deletion [27]. This score adheres to a discrete numerical scale: a score of −2 corresponds to a profound loss, suggestive of a potential homozygous deletion; a score of −1 signifies a shallow loss, possibly indicating a heterozygous deletion; a score of 0 is indicative of a diploid state; conversely, a score of 1 implies a low-level gain, often manifesting as a minor increase in copy number, which may encompass broad regions and could represent a heterozygous amplification; and lastly, a score of 2 signifies a pronounced amplification, characterized by a substantial increase in copy number, typically concentrated within focal regions, potentially reflective of a homozygous amplification.

#### 2.3.3. SNV, CNV and Survival

Following the SNV analysis, a survival analysis was conducted employing the R package survival (https://CRAN.R-project.org/view=Survival) (accessed on 5 June 2024) while using Cox Proportional-Hazards modeling and Logrank tests to assess survival disparities between the wild-type (WT) and mutant groups. All cancers and survival types with a significant Cox *p*-value (<0.05) were individually plotted using Kaplan–Meier curves.

Regarding the CNV survival analysis, merged CNV data and clinical survival data were categorized into WT, Amplification, and Deleterious groups, and a survival analysis was conducted using the same method as for SNV survival. All groups, including those with fewer than two samples, were retained for the survival curve visualization.

#### 2.3.4. Methylation and Survival

The study also utilized Illumina HumanMethylation 450k level 3 data from the TCGA database to assess methylation and survival [28]. We focused on fourteen (14) cancer types, each having a minimum of ten paired tumor and adjacent non-tumor samples. A correlation analysis was performed to identify the methylation site that most negatively correlated with gene expression. Clinical data from 33 cancer types were obtained, excluding samples with competing cancer-related mortality risks (for DSS and DFI data). Methylation and survival data were categorized into high and low methylation groups based on the median methylation levels. A survival analysis, including Cox Proportional-Hazards modeling and Logrank testing, was conducted using the R (v4.1.1) package survival to assess survival differences between the groups.

### 2.4. Immunotherapy and Drug Sensitivity Analysis

#### 2.4.1. Immune Response

We used ROC Plotter (https://www.rocplot.org/) (accessed on 10 July 2024) [29] to analyze *GZMA*, *GZMB*, *GZMK* and *PRF1* mRNA expression levels in the context of their sensitivity to breast cancer therapies, including endocrine therapy, anti-HER2 therapy and chemotherapy. The following parameters were utilized: pathological complete response (*n* = 1775) with “any” selected for endocrine therapy, anti-HER2 therapy and chemotherapy. There were 64 patient samples that met the conditions for endocrine therapy (19 non-responders and 45 responders), 217 samples for anti-HER2 therapy (122 non-responders and 95 responders), and 1632 samples for chemotherapy (1100 non-responders and 532 responders).

We further analyzed the gene expression levels in the ovarian cancer therapy samples, looking at chemotherapy and targeted Avastin therapy. The following parameters were selected: pathological response (*n* = 1022) with “any” for chemotherapy and “Avastin” selected for targeted therapy. There were 1022 patient samples for chemotherapy (235 non-responders and 787 responders) and 47 patient samples for Avastin targeted therapy (12 non-responders and 35 responders). The Mann–Whitney test and *p*-values were utilized for all of the analyses.

ROC plotter was also used to investigate the association between *GZMA*, *GZMB*, *GZMK* and *PRF1* and overall immunotherapy response. The samples used in the analysis were obtained before any treatment (pre-treatment), and the study considered individuals who had undergone any form of immune checkpoint inhibitor therapy, with an interest in understanding the gene’s role in treatment response. Furthermore, the analysis was not restricted to a specific tissue type, encompassing samples from various tissue origins. These settings were chosen to explore the predictive performance of the genes involved in the immune response in the context of immune checkpoint inhibitor therapy.

When examining the genes *GZMA*, *GZMB*, *GZMK* and *PRF1* within ROC Plotter, the combined response parameter was selected (*n* = 1434) along with “any immune checkpoint inhibitor therapy” for treatment type. We assessed their involvement in treatment responses using a dataset of 1267 samples, comprising 809 non-responders and 458 responders to any immune checkpoint inhibitors therapy. Filters were applied to select pretreatment samples, all genders, primary tumors and metastatic cases. Tissue of origin, excluding outliers, displaying quartiles and utilizing a multi-gene approach were among the settings employed for a comprehensive evaluation of *GZMA*’s relevance in immune checkpoint inhibitor responses.

Tumor Immune Dysfunction and Exclusion (TIDE) (http://tide.dfci.harvard.edu) (accessed on 25 June 2024) was also utilized to analyze the correlation between *GZMA*, *GZMB*, *GZMK* and *PRF1* mRNA expression and the outcomes of various immunotherapies [30,31].

#### 2.4.2. Drug Sensitivity and Therapeutic Response

The analyses presented in this study leveraged data available on the Genomics of Drug Sensitivity in Cancer (GDSC) website [32]. The GDSC database provided information on the half-maximal inhibitory concentration (IC50) values for 265 small molecules across a collection of 860 different cancer cell lines, and these were then correlated with the mRNA gene expression profiles of the genes of interest using Pearson correlation analysis. Subsequently, *p*-values were adjusted using the FDR method. This approach enabled the identification of significant correlations between gene expression patterns and drug responses, offering valuable insights into potential therapeutic targets and the mechanisms underlying drug sensitivity.

Additionally, we gathered data from the Genomics of Therapeutics Response Portal (CTRP) [33], including the IC50 values for 481 small molecules across 1001 different cell lines, along with their respective mRNA gene expression profiles. These datasets were then combined, and a Pearson correlation analysis was also carried out to establish the relationships between gene mRNA expression and drug IC50 values. *p*-values were adjusted using the FDR.

## 3. Results

### 3.1. GZMA, GZMB, GZMK and PRF1 Gene Expression in Multiple Cancers

The expression of *GZMA*, *GZMB*, *GZMK* and *PRF1* was assessed in multiple cancers. *GZMB* and *PRF1* showed consistently elevated expression in tumor samples compared to normal tissues across several cancers, such as BRCA and KIRC, suggesting their role in tumor immune activity. However, certain cancers, like UCEC and LUAD, exhibited more modest differences, highlighting the heterogeneity in immune gene activity across cancers (Figure 1b). Focused boxplot comparisons of selected cancers are provided in Figure 1c, which emphasize the specific statistical differences (e.g., between KIRC, BRCA and LUAD). Such a targeted analysis delved deeper into the significant trends, allowing for a clearer interpretation of the gene-specific differential expression pattern. The highest *GZMA* mRNA levels were observed in KIRC (*p* = 3.9 × $10^{-10}$), BRCA (*p* = 1.08 × $10^{-3}$) and KIRP (*p* = 4.27 × $10^{-3}$) compared to their adjacent normal tissues. A significant decrease in the expression of *GZMA* was found in COAD (*p* = 4.69 × $10^{-3}$) and LUSC (*p* = 8.25 × 10^−3^). Additionally, *GZMB* expression was highest in KIRC (*p* = 2.75 × $10^{-11}$), COAD (*p* = 2.28 × $10^{-4}$) and BRCA (*p* = 4.65 × $10^{-4}$), whereas, its lowest levels were found in LUSC, THCA and LUAD. *GZMK* was significantly upregulated in BRCA (8.02 × 10^−5^) and KIRC (7.27 × 10^−8^), and downregulated in LIHC (8.11 × 10^−3^). Furthermore, *PRF1* mRNA expression was significantly increased in KIRC (*p* = 2.42 × $10^{-15}$), HNSC (*p* = 4.38 × $10^{-3}$) and KICH (*p* = 0.02) and significantly decreased in LUSC (*p* = 1.13 × $10^{-13}$) and LUAD (*p* = 7.06 × $10^{-10}$). Overall, kidney and breast cancers displayed the highest expression of *GZMA*, *GZMB*, *GZMK* and *PRF1* (Figure 1a–c).

To determine the prognostic implications of *GZMA*, *GZMB*, *GZMK* and *PRF1* across the different cancer types, we explored their Kaplan–Meier plots. The greatest significance, as determined by the Cox *p*-value, was observed in SKCM for all the genes of interest. High *GZMA*, *GZMB*, *GZMK* and *PRF1* expression correlated with better overall survival (OS), progression-free survival (PFS) and disease-specific survival (DSS) in SKCM (Logrank, *p* < 0.01) (Figure 2a).

In contrast, in UVM, higher expression of *GZMB* was linked to a substantially higher risk of death and extremely low CoxP values for both DSS and PFS. Additionally, in LIHC, lower expression of *PRF1* was associated with a higher risk of death and low CoxP values for DFI (disease-free interval). These findings suggest that the expression of the granzyme family of genes and perforin is closely related to survival outcomes.

We next employed GEPIA2 to determine the correlations between *GZMA*, *GZMB*, *GZMK* and *PRF1* and the signatures of exhausted T cells (*HAVCR2*, *TIGIT*, *LAG3*, *PDCD1*, *CXCL13* and *LAYN*) and resting (*FOXP3* and *IL2RA*) and effector regulatory T cells (Tregs) (*FOXP3*, *CTLA4*, *CCR8* and *TNFRSF9*) in a comprehensive selection of tumor types. The signatures of exhausted T cells, resting T regulatory cells and effector regulatory T cells were significantly increased (*p* < 0.01) in CESC, DLBC, GBM, KIRC, KIRP, PAAD, SKCM, STAD and TGCTs (Figure 2b), indicating a complex interplay between the immune system and cancer cells. The increased expression of signatures of exhausted T cells and resting and effector Tregs in tumors compared to normal tissues highlights several key points about the tumor–immune cell interaction. Exhausted T cells are T cells that have been chronically exposed to antigens and exhibit decreased function, including reduced proliferation and cytokine production. They often express inhibitory receptors, such as PD-1 or CTLA-4. Higher numbers of exhausted T cells in tumors suggests that the immune response has been ongoing, but is not effectively clearing the cancer cells. This state of exhaustion can be a mechanism through which tumors evade immune destruction [34].

In addition, Tregs are a subset of T cells that play a critical role in maintaining immune tolerance and preventing autoimmune disease. However, in the context of cancer, Treg cells can suppress the anti-tumor immune response, allowing cancer cells to grow unchecked. Depletion of Treg cells within the TME has been shown to increase anti-tumor effects [35]. Furthermore, effector Treg cells are a more active form of Treg cells that are directly engaged in suppressing effector T cell functions within the tumor microenvironment. Like resting Treg cells, their depletion has been shown to change the tumor microenvironment from a state of tumor immunity to attacking malignant cells [36]. This suggests that a robust mechanism of immune suppression is being employed by the tumor to protect itself from immune-mediated destruction.

### 3.2. Pathway Analysis

A pathway analysis was conducted to evaluate the inhibitory and activating effects of ten cancer-related pathways on the expression of *GZMA*, *GZMB*, *GZMK* and *PRF1*. This analysis encompassed 7876 samples from 33 cancer types within the GSCA database (Figure 3a). We observed a strong activation effect of GZMA mRNA expression on the Apoptosis (44%), Hormone ER (31%) and EMT (19%) pathways. Inhibitory effects for *GZMA* in cancer were seen mainly in the RTK (19%) pathway, with some inhibitory effects in the Hormone AR (9%), PI3K/AKT (9%), TSC/mTOR (9%), Cell Cycle (6%), and DNA Damage (6%) pathways. Similarly, *GZMB* showed stronger activation in the Apoptosis (56%), EMT (31%), Hormone ER (31%) and Cell Cycle (16%) pathways across the different cancer types, while inhibitory effects were seen in the RTK (22%), PI3K/AKT (16%), Hormone AR (12%), DNA Damage (12%), RAS-MAPK (9%) and TSC/mTOR (9%) pathways. *GZMK* was also significantly involved in the activation of the Hormone ER (34%), EMT (28%) and Apoptosis (25%) pathways, and it was implicated in the inhibition of the RTK (22%) and Cell Cycle (16%) pathways. Finally, *PRF1* displayed the greatest activating effects on the Apoptosis pathway (56%), followed by the EMT (31%) and Hormone ER (28%) pathways in cancer. In contrast, most of the inhibitory effects of *PRF1* were seen in the Hormone AR (16%), DNA Damage (12%), RTK (12%), PI3K/AKT (9%), Hormone ER (6%), and TSC/mTOR (6%) pathways. Regarding the Hormone_ER and AR signatures, these are particularly relevant to hormone-sensitive cancers, such as BRCA and PRAD (prostate adenocarcinoma), given their association with estrogen receptor (ER) and androgen receptor (AR) activity. For the other cancer types, these signatures may hold less significance depending on the tumor’s biology and reliance on hormone signaling pathways.

More specifically, in BRCA, higher mRNA expression levels of *GZMA*, *GZMB*, *GZMK* and *PRF1* were observed to activate the Apoptosis pathway, as well as the Hormone ER and EMT pathways (Figure 3b). This activation was expected due to their established roles in immune responses and cell death [37]. In summary, the pathway analysis revealed that *GZMA*, *GZMB*, *GZMK* and *PRF1* mRNA expression predominantly activated key cancer-related pathways, including the Apoptosis, EMT and Hormone ER pathways in various cancer types, highlighting their roles in immune-mediated cell death and tumor microenvironment regulation. Conversely, these genes exhibited inhibitory effects primarily in pathways related to growth signaling and cell survival, such as the RTK, PI3K/AKT and TSC/mTOR pathways, suggesting potential tumor-suppressive functions.

In order to have a better understanding of the mRNA expression of *GZMA*, *GZMB*, *GZMK* and *PRF1*, we analyzed the correlation between expression and immune infiltration across 24 immune cells on the ImmuneCellAI platform (Figure 3c). We highlighted those with the most significant correlations among the 24 distinct immune cell types. Among the different tumor types, BRCA, BLCA, SKCM, ESCA, CHOL and ACC stood out. In particular, when examining cytotoxic T cells in the different cancers, we observed the highest correlation values for *GZMA* (correlation coefficient: 0.921), *GZMB* (correlation coefficient: 0.902) and *PRF1* (correlation coefficient: 0.928) in BLCA. Our analysis revealed significant correlations between *GZMA*, *GZMB*, *GZMK* and *PRF1* expression and cytotoxic T cell infiltrates across various cancer types, underscoring the crucial roles of these genes in modulating the tumor microenvironment (Appendix A).

### 3.3. Somatic Mutations

We then characterized the SNV mutation status of *GZMA, GZMB, GZMK* and *PRF1* in multiple cancers based on the mutation rate and the numbers of deleterious mutations (Figure 4a). The numbers within the cells represent the mutation frequency, calculated as the proportion of samples within each cancer type harboring at least one deleterious mutation in the specified gene. Deleterious mutations include missense mutations, nonsense mutations, frameshift insertions/deletions, splice site mutations and in-frame insertions/deletions. In total, 4.49% of SKCM (out of a 468 tumors), 3.95% of UCEC (sample size of 531) and 2.68% of READ tumors (sample size of 149) had *GZMA* mutations. Additionally, 21 *GZMA* deleterious mutations were present in SKCM, 21 in UCEC and 4 in READ tumors.

Regarding the *GZMB* mutation rates in the different cancer types, UCEC had 14 deleterious and 6 non-deleterious mutations in 531 cases (2.64%). In SKCM, nine deleterious and five non-deleterious mutations were found in 468 cases (1.92%). UCS showed one deleterious mutation in 57 cases (1.75%).

Regarding the *GZMK* mutation rates, SKCM had 18 deleterious and 6 non-deleterious mutations in 468 cases (3.85%). UCEC showed 16 deleterious and 11 non-deleterious mutations in 531 cases (3.01%), while COAD had 9 deleterious mutations in 407 cases (2.21%).

For *PRF1* mutations, SKCM exhibited 24 deleterious and 15 non-deleterious mutations in 468 cases (5.13%). UCEC had 24 deleterious and 16 non-deleterious mutations in 531 cases (4.52%). STAD showed 17 mutations (3 non-deleterious) in 439 cases (3.87%), and COAD had 11 deleterious and 6 non-deleterious mutations in 407 cases (2.70%). These findings emphasize the considerable variability in *PRF1* SNV mutations across different cancer types (Figure 4a). Globally, there was an elevated prevalence of mainly missense SNVs in SKCM, UCEC and STAD across all four genes (Figure 4b).

Following the SNV analysis, we examined the proportion of heterozygous and homozygous mutations (both deletions and amplifications) in *GZMA*, *GZMB, GZMK* and *PRF1* in the different cancers. When looking at the CNV percentages in each cancer type, there were generally more heterozygous amplifications and deletions present compared to heterozygous amplifications and deletions (Figure 4c). The analysis of the CNVs in *GZMA* revealed that ACC exhibited the highest percentage of heterozygous amplifications (Hete. amp.) at 66.67%, followed by KIRC and LIHC. The examination of the deletions in *GZMA* revealed that LUSC exhibited the highest percentage of total deletions (Total del.) at 71.26%, with a significant proportion attributed to heterozygous deletions (Hete. del.) at 70.06%. TGCTs and ESCA also demonstrated substantial rates of total deletions, with significant contributions from both heterozygous and homozygous deletions. These findings underscore the prevalence of deletions in *GZMA* across various cancer types, particularly in LUSC.

Within this analysis, TGCTs were the highest in terms of total amplifications (Total amp.) at 33.33%, primarily due to heterozygous amplifications (Hete. amp.: 33.33%). LUAD and KICH also had high rates of total amplifications, accredited to heterozygous amplifications. The *GZMB* CNV analysis found that UCS had the highest percentage of total deletions (Total dele.) at 50.00%, denoting the prevalence of this type of variation in UCS, which was primarily composed of heterozygous deletions (Hete. del.: 50%), which highlights the importance of heterozygous deletions in UCS. Other cancer types, such as MESO, READ and KIRC, also demonstrated noteworthy percentages of total deletions, with a significant proportion being a result of heterozygous deletions. These findings stress the importance of *GZMB* CNV deletions, particularly heterozygous deletions, in UCS (50%), MESO (42.53%), READ (40%) and KIRC (39.20%).

The analysis of *GZMK* found that ACC had the highest percentage of amplifications at 68.89%, with mostly heterozygous amplifications (66.67%). KIRC and LIHC had the next greatest values for amplification of *GZMK* at 31.06% and 28.38%, respectively. For SARC, when looking at *GZMK*, the data showed a total amplification rate of 27.63% and a total deletion rate of 15.18%. Heterozygous amplifications and deletions were observed at rates of 26.85% and 15.18%, respectively, while homozygous amplifications constituted 0.78%. In LUAD, a total amplification rate of 25.39% with a total deletion rate of 31.40% were recorded.

The pan-cancer CNV analysis of *PRF1* indicated that the percentage of total amplifications was highest in UCS (42.86%). Heterozygous amplifications (Hete. amp.) were highest, also at 42.86%. This demonstrates the commonness of heterozygous amplifications in UCS. Other cancer types, such as ACC, OV and UCEC, displayed significantly high percentages of total amplifications, with a larger fraction attributable to heterozygous amplifications. This underscores the significance of PRF1 CNV amplifications, particularly heterozygous amplifications, in UCS (Total amp.: 42.86%), ACC (28.89%), OV (22.80%) and UCEC (20.96%). On the other hand, GBM exhibited the highest percentage of total deletions (87.35%), primarily comprising heterozygous deletions (Hete. del.) at a slightly lower percentage (87%). KICH, SKCM, SARC and TGCTs also demonstrated substantial percentages of total deletions, with notable proportions attributed to heterozygous deletions. Specifically, KICH displayed a total deletion rate of 72.73%, while SKCM, SARC, and TGCTs exhibited rates of 60.49%, 54.47% and 50.67%, respectively; these deletions were predominantly composed of heterozygous deletions. In summary, the CNV analysis of GZMA, GZMB, GZMK and PRF1 across multiple cancer types highlighted a remarkable prevalence of heterozygous amplifications and deletions compared to homozygous variations. GZMA deletions were most prominent in LUSC, while TGCTs showed the highest amplification rates. For GZMB, heterozygous deletions were dominant in UCS, MESO, READ and KIRC, emphasizing their significance in these cancers. Similarly, GZMK exhibited high amplification rates in ACC and KIRC, with deletions also observed in cancers like SARC and LUAD.

To further explore the impact of the SNV and CNV mutations in *GZMA*, *GZMB*, *GZMK* and *PRF1* in the different cancers, we assessed the survival differences between mutant and wild-type forms of the genes (Figure 5a–c). In BRCA, mutations in *GZMA* were associated with a higher risk of death in terms of both DSS and OS. Conversely, in UCEC, mutations in *GZMA* were linked to a lower risk of death in terms of progression-free survival (PFS). The survival analysis focusing on SNV mutations in *GZMB* across the various cancers demonstrated a significant association in GBM for PFS. When looking at the association between *GZMK* SNV mutations and survival outcomes, significant associations were seen in BRCA. Mutations in *GZMK* were linked to a higher risk of death in terms of DSS, OS and PFS. The DSS and OS of *GZMK* SNV mutations in LUAD revealed significant survival association outcomes as well. The analysis of SNV mutations in *PRF1* across the different cancer types revealed significant associations with survival outcomes in COAD and UCEC. In COAD, mutations in *PRF1* were significantly associated with a higher risk of death and shorter PFS. In UCEC, a significant association with PFS was observed, indicating a distinct survival pattern.

Additionally, we analyzed CNV mutations in *GZMA* across various cancer types to delve deeper into the genomic alterations that may contribute to cancer survival outcomes (Figure 5c). The most significant survival associations were observed in KIRP, THCA, thymoma, KICH, UCEC and KIRC, as well as SARC.

Regarding CNV mutations in *GZMB*, the most significant survival associations were observed in Mesothelioma (MESO), Lower-Grade Glioma (LGG), Kidney Renal Papillary Cell Carcinoma (KIRP) and Acute Myeloid Leukemia (LAML), along with Kidney Renal Clear Cell Carcinoma (KIRC).

When looking at *GZMK* in the different cancers, it was found that in KIRP, it was significantly associated with DSS, OS and PFS. Additionally, in THCA, *GZMK* was significantly correlated with DSS and in THYM, it was significantly associated with PFS.

Similar, we found important patterns for *PRF1* CNVs (Figure 5c). In particular, for LGG, the impact on OS, DFI and DSS was profound. Additionally, in LAML, the association with OS remained noteworthy. Furthermore, MESO exhibited significant associations between both DSS and OS and *PRF1* CNVs, highlighting their relevance in predicting survival outcomes across different cancer types. The analysis of the Kaplan–Meier plots for the most significant cancer association (LGG) with *PRF1* gene CNV mutations revealed that in OS, PFS and DSS, deletion mutations were associated with adverse survival outcomes when compared to wild type. These findings underscore the diverse impact of mutations in these immune-related genes on survival outcomes across different cancer types. The analysis of copy number variation (CNV) mutations in *GZMA*, *GZMB*, *GZMK* and *PRF1* across the various cancer types revealed significant associations with survival outcomes, highlighting their potential prognostic value. *GZMA* CNVs were notably associated with survival in multiple cancers, including KIRP, THCA, thymoma, KICH, UCEC, KIRC and SARC. *GZMB* CNVs exhibited strong correlations with survival in MESO, LGG, KIRP, LAML and KIRC. *GZMK* CNVs were significantly linked to survival metrics such as DSS, OS and PFS in KIRP, THCA and THYM.

As a comprehensive pan-cancer exploration, we next investigated the impact of DNA methylation of *GZMA*, *GZMB*, *GZMK* and *PRF1* on survival outcomes. In LGG, *GZMA* hypomethylation exhibited a consistently lower risk of death (DSS and OS) and enhanced PFS. Conversely, in UVM, lower methylation at the same locus was associated with markedly increased OS and a decreased risk of death (DSS). Additionally, alterations in *GZMA* methylation levels were linked to survival disparities in ACC and SKCM, further emphasizing the potential prognostic implications of DNA methylation in the context of *GZMA* across diverse cancer types.

Furthermore, in LGG, lower *GZMB* methylation correlated with improved OS and DSS, and enhanced PFS. Similarly, in KIRP, lower *GZMB* methylation was associated with improved PFS and a decreased risk of death. Conversely, in CESC, higher *GZMB* methylation correlated with an increased DFI. In UVM, lower *GZMB* methylation was linked to improved OS.

When looking at *GZMK*, in UVM, higher methylation levels were associated with a reduced risk of death in terms of both PFS and OS. Similarly, in MESO, higher *GZMK* methylation was linked to a decreased risk of death in terms of PFS.

Finally, lower *PRF1* methylation in LGG correlated with improved OS and DSS and enhanced PFS. On the contrary, higher *PRF1* methylation was associated with reduced PFS in LGG. In SKCM, higher *PRF1* methylation was linked to an increased risk of death (DSS) and reduced OS. Additionally, higher *PRF1* methylation in HNSC correlated with an elevated risk of death (DSS). This investigation highlighted the diverse associations between DNA methylation of *GZMA*, *GZMB*, *GZMK*, and *PRF1* and survival outcomes, displaying the potential prognostic implications of epigenetic modifications in immune-related genes across different cancer types.

### 3.4. Prediction of Sensitivity to Immunotherapy and Other Drugs

In order to understand *GZMA*, *GZMB*, *GZMK* and *PRF1* expression in the context of responses to immunotherapy and other therapeutic approaches, we used ROC plotter. Interestingly, all four genes showed significant sensitivity to immunotherapy (*p* < 0.05, Mann–Whitney test), as they were downregulated in the responders to immunotherapy compared to non-responders (Figure 6a).

Focusing on breast cancer, we explored each gene’s expression according to patient sensitivity to different therapies, including endocrine therapy, anti-HER2 therapy and chemotherapy. *GZMA*, *GZMB* and *GZMK* mRNA levels were all upregulated in the responders to chemotherapy. Additionally, *PRF1* gene expression was upregulated in the responder group for anti-HER2 therapy, showing significant sensitivity to this therapy (Figure 6b). Therefore, the expression analysis of *GZMA*, *GZMB*, *GZMK* and *PRF1* highlighted their relevance to therapeutic responses. The expression of these genes may serve as potential biomarkers for predicting sensitivity to specific therapeutic approaches, including immunotherapy and targeted treatments for breast cancer.

We also explored the differences in the expression between responders and non-responders to chemotherapy and Avastin targeted therapy (Figure 7a). Of note, *GZMB* expression was significantly higher in ovarian cancer patients that were sensitive to chemotherapy. *GZMK* expression was also significantly higher among responders to Avastin therapy.

Additionally, we collected the IC50 values of drugs from various cancers and cell lines from the GDSC (Figure 7b) and CTRP (Figure 7c) databases in order to correlate them with *GZMA, GZMB, GZMK* and *PRF1* expression levels.

Notably when looking at the GDSC data (Figure 7b), *GZMA* expression was negatively correlated with sensitivity to Methotrexate (Pearson’s rho = −0.35, FDR = 5.87 × $10^{-24}$), CH5424802 (Pearson’s rho = −0.32, FDR = 2.81 × $10^{-20}$), XMD14-99 (Pearson’s rho = −0.31, FDR = 1.54 × $10^{-20}$), CP466722 (Pearson’s rho = −0.29, FDR = 1.09 × $10^{-17}$), TPCA-1 (Pearson’s rho = −0.27, FDR = 6.05 × $10^{-16}$), MS-275 (Pearson’s rho = −0.27, FDR = 1.11 × $10^{-4}$) and Vorinostat (Pearson’s rho = −0.26, FDR = 8.77 × $10^{-14}$). In addition, *GZMA* expression was positively correlated with sensitivity to 17-AAG (FDR = 2.79 × $10^{-5}$) and Docetaxel (FDR = 4.70 × $10^{-4}$. These correlations suggest potential associations between *GZMA* expression and the response to these specific drugs.

Further, *GZMB* was negatively correlated with sensitivity to CH5424802 (Pearson’s rho = −0.57, FDR = 6.61 × $10^{-78}$), Crizotinib (Pearson’s rho = −0.41, FDR = 5.81 × $10^{-14}$), TAE684 (Pearson’s rho = −0.40, FDR = 1.05 × $10^{-13}$), XMD14-99 (Pearson’s rho = −0.39, FDR = 4.66 × $10^{-33}$), MPS-1-IN-1 (Pearson’s rho = −0.35, FDR = 2.34 × $10^{-25}$), XMD15-27 (Pearson’s rho = −0.26, FDR = 3.80 × $10^{-13}$) and Temozolomide (Pearson’s rho = −0.25, FDR = 5.13 × $10^{-11}$). We also found weaker positive correlations between *GZMB* and sensitivity to Dasatinib (FDR = 0.04) and Tapsigargin (FDR = 0.04).

Notably, *PRF1* was negatively correlated with sensitivity to CH5424802 (Pearson’s rho = −0.54, FDR = 9.55 × $10^{-68}$), XMD14-99 (Pearson’s rho = −0.43, FDR = 3.33 × $10^{-41}$), Crizotinib (Pearson’s rho = −0.38, FDR = 1.32 × $10^{-11}$), MPS-1-IN-1 (Pearson’s rho = −0.36, FDR = 3.79 × $10^{-27}$), XMD15-27 (Pearson’s rho = −0.34, FDR = 2.46 × $10^{-22}$) and Methotrexate (Pearson’s rho = −0.34, FDR = 7.11 × $10^{-23}$). It was also positively correlated with sensitivity to KIN001-135 (FDR =: 0.09), GSK1904529A (FDR = 3.63 × $10^{-3}$) and Docetaxel (FDR = 3.33 × $10^{-3}$).

Similarly, *GZMK* was negatively correlated with sensitivity to Methotrexate (Pearson’s rho = −0.21, FDR = 2.26 × $10^{-9}$), TPCA-1 (Pearson’s rho = −0.20, FDR = 4.91 × $10^{-9}$), PIK-93 (Pearson’s rho = −0.17, FDR = 7.16 × $10^{-7}$), CP466722 (Pearson’s rho = −0.14, FDR = 3.49 × $10^{-5}$) and SNX-2112 (Pearson’s rho = −0.14, FDR = 1.12 × $10^{-4}$).

Regarding drugs in the CTRP dataset (Figure 7c), *GZMA* exhibited significant negative correlations with sensitivity to neopeltolide (FDR = 3.46 × $10^{-5}$), GSK461364 (FDR = 1.45 × $10^{-14}$) and Manumycin A (FDR = 1.85 × $10^{-14}$). Significant positive correlations were seen between *GZMA* and sensitivity to Abiraterone (FDR = 0.03), Austocystin (FDR = 4.15 × $10^{-5}$) and VAF-347 (FDR = 6.61 × $10^{-3}$). *GZMB* displayed a significant negative correlation with sensitivity to CR-1-31B (FDR = 1.26 × $10^{-6}$), MLN2238 (FDR = 3.13 × $10^{-6}$) and NSC632839 (FDR = 2.46 × $10^{-6}$) and a positive correlation with sensitivity to SB-525334 (FDR = 0.05).

Furthermore, *GZMK* was negatively correlated with sensitivity to Bosutinib (Pearson’s rho = −0.12, FDR = 2.27 × $10^{-3}$), NSC23766 (Pearson’s rho = −0.11, FDR = 5.20 × $10^{-3}$), BRD6340 (Pearson’s rho = −0.11, FDR = 6.26 × $10^{-3}$), elocalcitol (Pearson’s rho = −0.11, FDR = 6.65 × $10^{-3}$) and Fluorouracil (Pearson’s rho = −0.10, FDR = 8.39 × $10^{-3}$).

*PRF1* was negatively correlated with sensitivity to Tozasertib (FDR = 0.05), PF-3758309 (FDR = 1.71 × $10^{-4}$), belinostat (FDR = 6.24 × $10^{-4}$) and Neopeltolide (FDR = 0.04) and positively correlated with sensitivity to Simvastatin (FDR = 9.33 × $10^{-3}$). Collectively, these findings underscore the potential of *GZMA*, *GZMB*, *GZMK* and *PRF1* as promising indicators for predicting drug responses.

Finally, we used TIDE’s regulator prioritization to explore the relationship between *GZMA*, *GZMB*, *GZMK* and *PRF1* expression and immunosuppression (Figure 7d). All of these genes were associated with multiple indicators of immunosuppression, including T cell dysfunction values and ICB response outcomes across various patient cohorts. TIDE was also utilized to depict the relationship between gene expression and effectiveness of various therapies from various patient cohorts in clinical trials that involved immune checkpoint blockade (Table 2, Table 3, Table 4 and Table 5). In the context of glioblastoma, “post” indicates that these patients received immune checkpoint blockade therapy after experiencing disease progression following prior treatments. Conversely, “pre” indicates that the therapy was administered prior to disease recurrence. Notably, GZMA showed high positive correlations with kidney cancer in the context of ICB therapy, suggesting its potential as a predictive biomarker for therapeutic outcomes. Similarly, *GZMB* and *GZMK* were most strongly correlated with melanoma, underscoring the relevance of these genes in mediating anti-tumor immune responses in this cancer type. Furthermore, bladder cancer demonstrated the highest positive correlation with OS, while glioblastoma showed the highest correlation with PFS. The analysis using TIDE’s regulator prioritization revealed significant associations between the expression of *GZMA*, *GZMB*, *GZMK* and *PRF1* and various indicators of immunosuppression, including T cell dysfunction and immune checkpoint blockade (ICB) response across multiple patient cohorts.

## 4. Discussion

Our findings align with those of recent studies emphasizing the significance of immune-related genes, specifically *GZMA*, *GZMB*, *GZMK* and *PRF1*, in cancer pathogenesis and clinical outcomes [8,38,39,40,41,42,43]. The mutation analysis revealed distinct mutation patterns in these genes in many cancers, consistent with the idea that the immune system’s role in cancer is highly context-dependent and varies across different cancer types. Research has shown improved prognosis and overall survival for the TMB-high melanoma patients when treated with immune checkpoint inhibitors (ICIs), indicating a possible interplay between immune system dysfunction and mutation rate [44]. Similar trends have been observed in other cancer types, here TMB-high status correlates with enhanced responses to ICIs and better clinical outcomes [45,46].

Our study’s focus on CNVs aligns with recent research highlighting the importance of genomic alterations in immune-related genes. There is evidence of genetic mutations related to downregulation of *GZMA* and *PRF1*, leading to immune evasion in LUSC [47]. Moreover, our results showing the importance of *GZMA* in breast cancer is further supported by research showing better survival rates among patients with high *GZMA* expression [40]. On the other hand, a recent transcriptomics analysis revealed the pro-proliferative role that extracellular *GZMA* plays in colorectal cancer progression. The experiment prevented cancer progression and gut inflammation when *GZMA* was pharmacologically inhibited [48]. Thus, further analysis is warranted to understand the effects of not only *GZMA* as a whole, but the effects of intracellular versus extracellular *GZMA* and its effects in cancer and the immune response. The contradictory impact of *GZMA* in these two types of malignancies supports the existing research investigating whether GZMA has a solely cytotoxic role in biological functioning [11].

Interestingly, our results for UCEC linked *GZMA* mutations to a lower risk of death in terms of PFS. The analysis of SNV mutations in *PRF1* across the different cancer types revealed significant associations with survival outcomes in UCEC as well. This was further supported by the investigation assessing the effects of *GZMA* downregulation in numerous cancers, including UCEC. It was shown that *GZMA* downregulation was associated with poor prognoses, which further implicates *GZMA* in tumor growth. Future research is imperative to understand the full scope of *GZMA* function in order to refine our approach to make it a therapeutic treatment target [49]. In regard to *PRF1*, while we believe our data look promising and in line with other research implicating *PRF1′s* role in poor prognosis in many cancer types, including UCEC, further research is warranted to determine how this protein impacts each cancer individually and the different functions it has [50].

Moreover, the diverse CNV patterns in *GZMB* across the different cancer types, such as amplifications in TGCTs and deletions in UCS, are consistent with the notion that *GZMB* alterations may have distinct implications in different cancer contexts [8]. Additionally, *GZMB* levels varied among the cancer types. Several studies have demonstrated a positive role of *GZMB* in suppressing tumor activity [51,52]. Yet, other data present *GZMB* as a pro-tumor player. In one study researching oral squamous cell carcinoma, increased numbers of *GZMB*-expressing Tregs were found in comparison with control patients [8,53]. Cancers such as Head and Neck Squamous Cell Carcinoma, Renal Clear Cell Carcinoma, and Stomach Adenocarcinoma have higher *GZMB* expression levels compared to lower-expressing *GZMB* cancers such as Lung Squamous Cell Carcinoma and Lung Adenocarcinoma [39]. Furthermore, studies have shown that low levels of *GZMB* expression were associated with a better probability of survival in certain cancers [8,54]. One possible cause of these variations is epigenetics as research has found that *GZMB* expression can vary due to epigenetic changes. Variations in *GZMB* transcription, differences in the levels and activity of transcription and translation repressors, and posttranscriptional modifications of E-Polymerase II have been observed [55]. These warrant further investigation given the limited research on the impact of epigenetics and its subdivisions within the context of different malignancies and *GZMB.*

The association between *GZMA* mutations and survival outcomes in BRCA aligns with studies emphasizing the role of immune-related genes in breast cancer prognosis. For example, research analyzing patients with triple-negative breast cancer showed that those with high *GZMB* expression levels had greater overall survival rates in addition to increased recurrence-free survival [56]. However, this study had ensured PD-L1 positivity in the patient cohort studied; thus, future research can look at PD-L1-negative patients for comparison. Moreover, *GZMK* expression was found to be significantly correlated with survival rates in breast cancer in another study [57]. These findings not only support the results of this study, but further support ongoing research demonstrating *GZMK*’s role in breast cancer progression. *GZMB* mutations in GBM and *PRF1* mutations in colorectal cancer align with the literature, suggesting an impact of these genes on patient survival in specific cancer types; meanwhile, a study on UCEC did not find a survival association [58,59,60,61,62].

Regarding our results from analyzing *GZMK* in Clear Cell Renal Cell Carcinoma, previous studies have also found higher expression of *GZMK*, with indications of T cell exhaustion [63,64]. There was also another study that supported the exploration of *GZMK* as a therapeutic target [64]. Prior research has similarly exhibited *GZMK*’s role in the anti-tumor response in melanoma, as it prohibited cell proliferation and migration. However, *GZMK* was found to not influence apoptosis in the bioinformatics analysis that was performed [65]. This finding warrants further investigation into why this granzyme exhibits stronger anti-tumor properties in certain contexts compared to others.

We also need to acknowledge that our pan-cancer analysis has some limitations. Firstly, while our study uncovered correlations between gene expression and immune cell infiltration, it did not definitively establish direct causal impact on patient survival. Therefore, prospective research initiatives exploring these complex interactions are warranted. Additionally, subsequent experimentation and clinical application are necessary to determine the reliability and validity of this project given that the analysis results are correlative findings. Since our analysis was performed on bulk transcriptomic data, we need to address the issue of biological significance. As perforins and granzymes are mainly expressed by infiltrating cells, their levels are mainly dependent on the number of infiltrating cells and therefore on tumor purity. To address this, we used ESTIMATE analysis. In addition, our current understanding of the complete roles of *GZMA*, *GZMB*, *GZMK* and *PRF1* are still evolving and other factors may play a role in the varying expression levels of these genes in the context of the different clinical prognoses seen in these cancer types. Possible explanations include tumor hyper-secretion of these proteins, their roles beyond immune responses (such as autoimmune activity) and differences in how they affect the sensitivity to apoptosis in different cancer types due to epigenetic changes [8,66,67].

As cancer genomics and our knowledge of the role of the immune system in cancer research continue to grow and evolve, using multi-omics as an approach to investigate these topics will not only aid researchers in understanding the relationship between the immune response and cancer, but also establish new directions for cancer immunotherapy research (Figure 8).

## 5. Conclusions

In conclusion, our study underscores the multifactorial impact of *GZMA, GZMB, GZMK* and *PRF1* and their potential as biomarkers for personalized cancer immunotherapy. Our findings highlight the critical role of these genes in cancer pathogenesis, progression and clinical outcomes. Through a pan-cancer analysis, we identified distinct mutation, copy number variation and methylation patterns in these genes, emphasizing their context-dependent roles across various cancer types. Our findings demonstrate the prognostic relevance of immune gene alterations and their influence on survival and drug sensitivity, particularly in the context of immune checkpoint inhibitors.

## Figures and Tables

**Figure 1 biomedicines-13-00408-f001:**
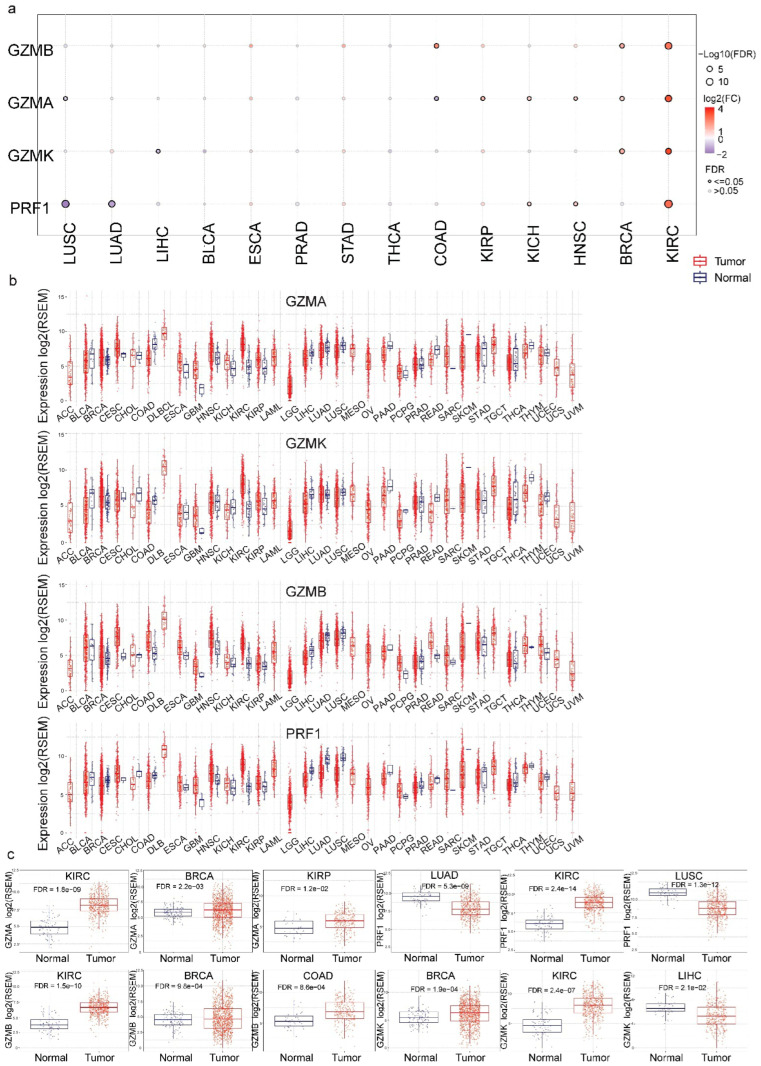
(**a**) The bubble plot depicts the differential expression of *GZMA*, *GZMB*, *GZMK* and *PRF1* (fold change and FDR) across 14 TCGA cancer types that had data available (LUSC, LUAD, LIHC, BLCA, ESCA, PRAD, STAD, THCA, COAD, KIRP, KICH, HNSC, BRCA and KIRC). (**b**) The boxplots depict the normalized log_2_(TPM + 1) expression levels of *GZMA*, *GZMB*, *GZMK* and *PRF1* across the 14 cancer types based on TCGA data. Statistical significance was assessed using the FDR-adjusted *p*-values for comparisons between tumor and normal tissues, which highlighted the differential expression patterns across cancers. (**c**) The boxplots depict *GZMA*, *GZMB*, *GZMK* and *PRF1* mRNA expression in specific cancers and their normal tissues.

**Figure 2 biomedicines-13-00408-f002:**
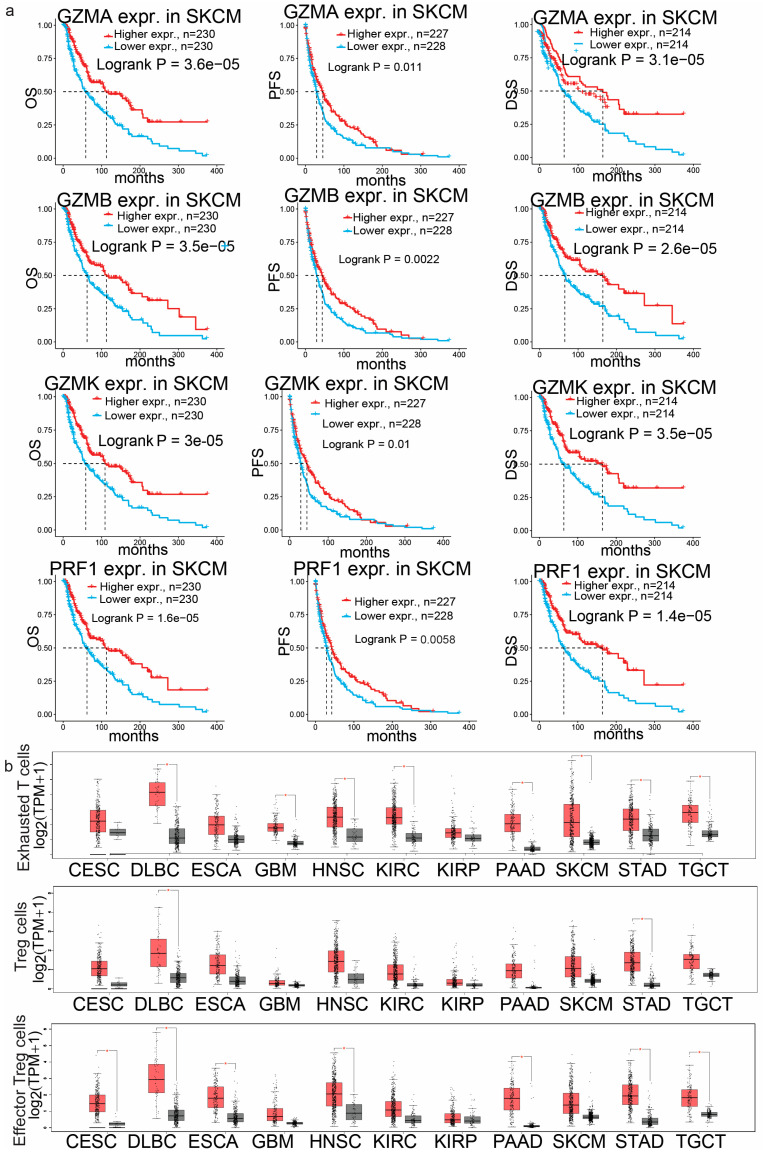
(**a**) The Kaplan–Meier plots display the survival differences (overall survival (OS), disease-specific survival (DSS) and progression-free survival (PFS)) between skin melanoma (SKCM) patients with low and high levels of *GZMA*, *GZMB*, *GZMK* and *PRF1* expression. (**b**) Boxplots showing the expression of molecular signatures specific for different T cell subtypes, including exhausted T cells (*HAVCR2*, *TIGIT*, *LAG3*, *PDCD1*, *CXCL13* and *LAYN*), resting regulatory T cells (*FOXP3* and *IL2RA*) and effector regulatory T cells (*FOXP3*, *CTLA4*, *CCR8* and *TNFRSF9*). Red boxes represent tumor tissues and gray boxes represent normal tissues. *, *p* < 0.05.

**Figure 3 biomedicines-13-00408-f003:**
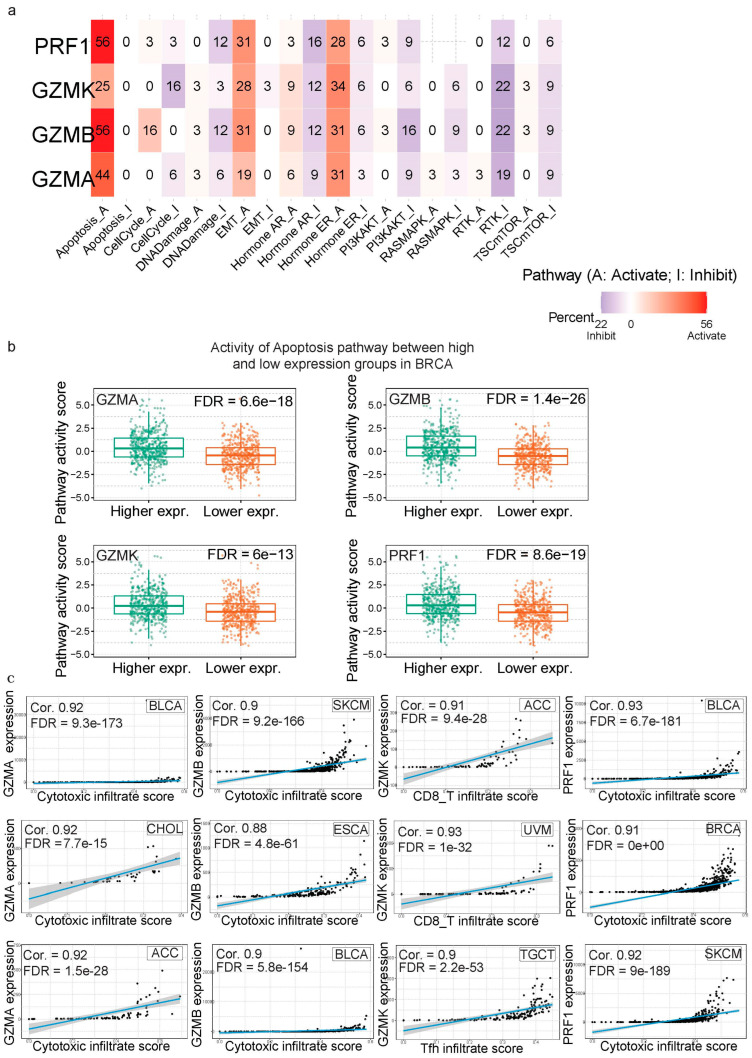
(**a**) Heatmap showing the percentage of cancer types showing activation or inhibition of 10 cancer-related pathways (Apoptosis, Cell Cycle, DNA Damage, EMT, Hormone AR, Hormone ER, PI3K/AKT, RAS/MAPK, RTK and TSC/mTOR pathways) by *GZMA*, *GZMB*, *GZMK* and *PRF1*. The numbers within the cells represent the percentage of cancers with significant pathway effects (FDR ≤ 0.05). The Hormone_ER and Hormone_AR signatures were most relevant in hormone-sensitive cancers, such as breast and prostate cancer. Blue shading indicates a shift towards inhibition, while red shading signifies a shift towards activation. (**b**) Boxplots depict the Apoptosis pathway activity scores of breast cancer (BRCA) samples with low and high expression of *GZMA*, *GZMB*, *GZMK* and *PRF1*. (**c**) Scatter plots illustrate the correlations between the mRNA expression of *GZMA*, *GZMB*, *GZMK* and *PRF1* and immune cell infiltrates; each plot displays a fitted line to visualize the relationship.

**Figure 4 biomedicines-13-00408-f004:**
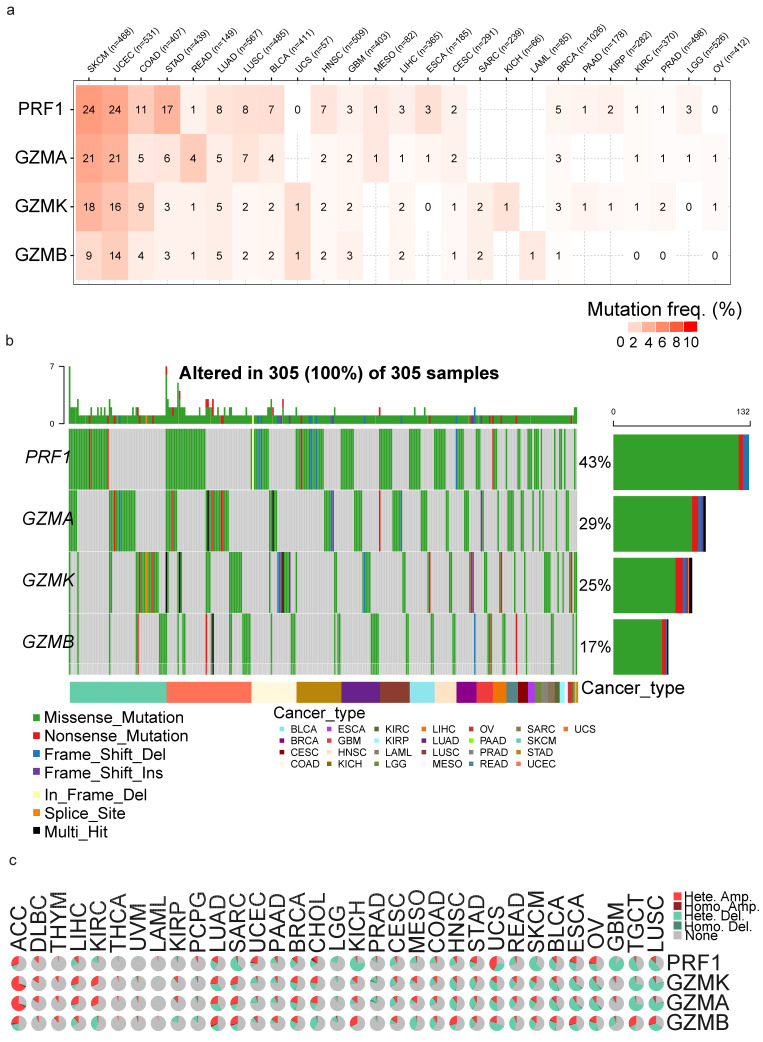
(**a**) Heatmap showing the frequency of deleterious single-nucleotide variations (SNVs) in *GZMA*, *GZMB*, *GZMK* and *PRF1* across various cancer types. The numbers within the cells indicate the percentage of tumors with deleterious SNVs. (**b**) Oncoplot displaying the mutational landscape of the *GZMA*, *GZMB*, *GZMK* and PRF1 genes across different cancer types. (**c**) The pie chart offers a comprehensive overview of the composition of the heterozygous and homozygous copy number variations (CNVs) for *GZMA, GZMB*, *GZMK* and *PRF1*. Each individual pie chart within the plot depicts the distribution of various CNV types for one specific gene within a particular cancer type. Distinct colors are employed to differentiate between the different CNV types.

**Figure 5 biomedicines-13-00408-f005:**
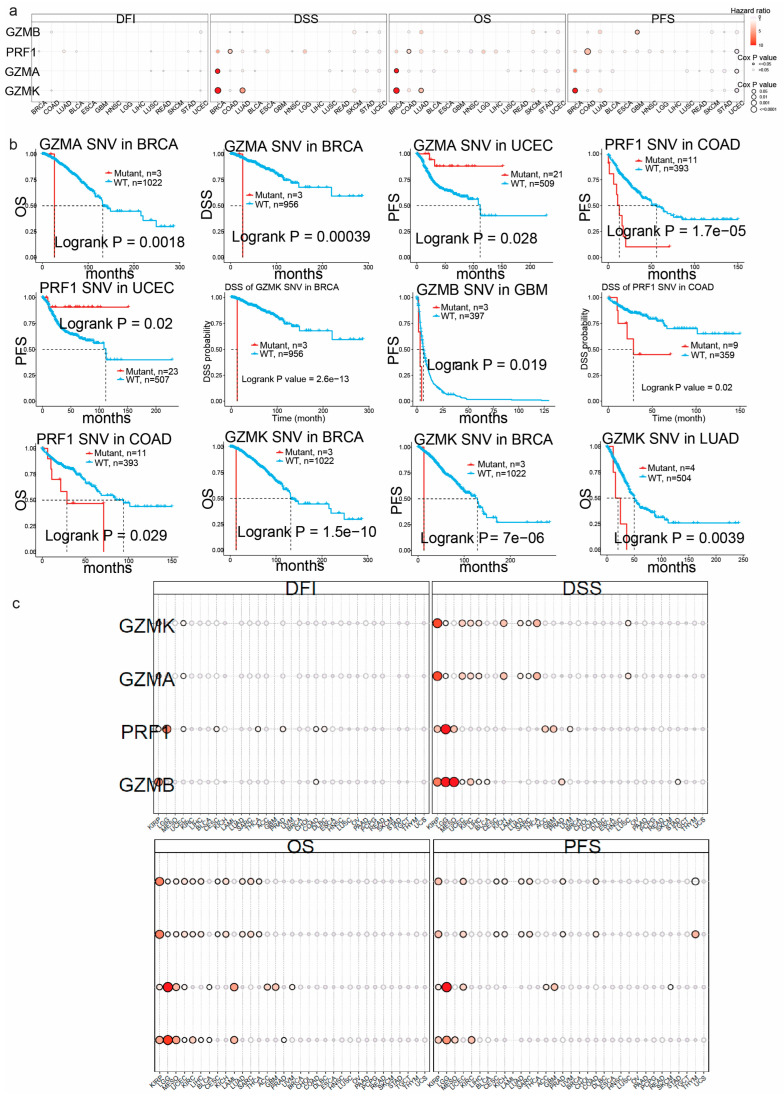
(**a**) Survival differences (DFI, DSS, OS and PFS) between CNV groups for *GZMA*, *GZMB, GZMK* and *PRF1* across different cancer types. Bubble color, transitioning from blue to red, denotes hazard ratios (low to high), while bubble size reflects the significance of Cox *p*-values. Notably, a black outline border signifies a Cox *p*-value of ≤0.05. (**b**) Associated Kaplan–Meier plots for gene and cancer types with SNV plotted for associations with *p*-values < 0.05. (**c**) Cancer survival disparities between CNV and wild type for *GZMA*, *GZMB*, *GZMK* and *PRF1*.

**Figure 6 biomedicines-13-00408-f006:**
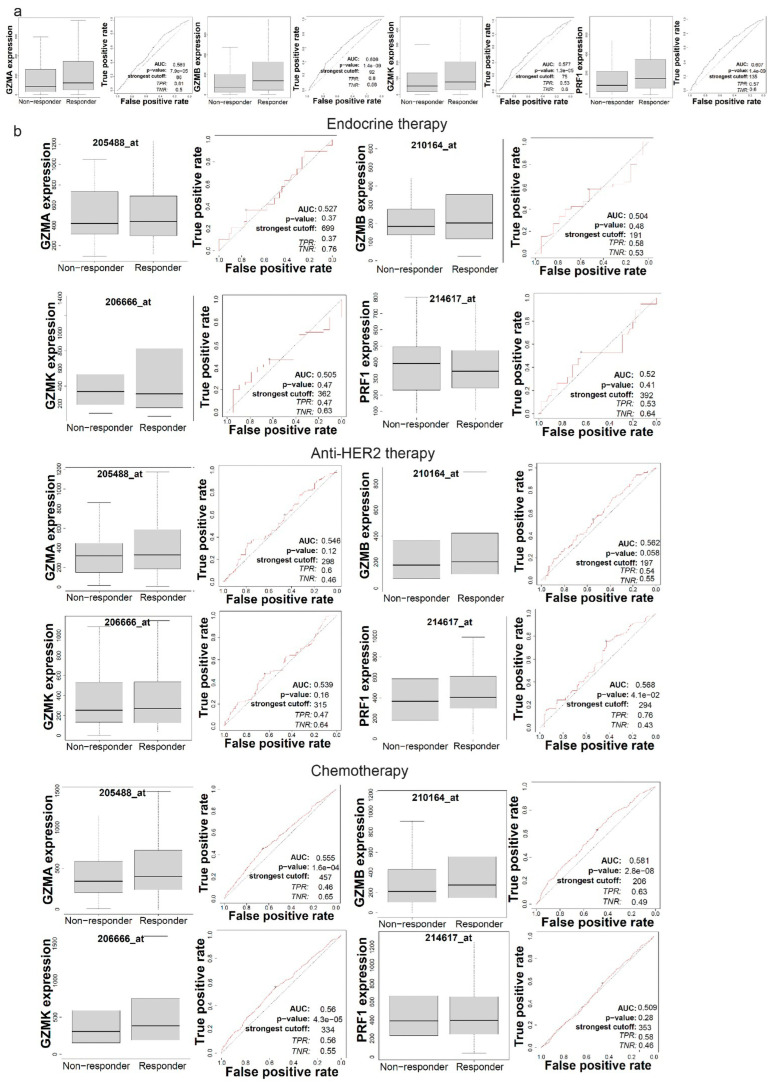
(**a**) Expression of *GZMA*, *GZMB*, *GZMK* and *PRF1* in responders and non-responders to immune checkpoint inhibitor therapies. (**b**) Expression of *GZMA*, *GZMB*, *GZMK* and *PRF1* in the context of sensitivity to endocrine therapy, anti-HER2 therapy and chemotherapy in breast cancer in terms of a complete response. ROC plotter was utilized for the analyses. A *p* < 0.05 indicates statistical significance.

**Figure 7 biomedicines-13-00408-f007:**
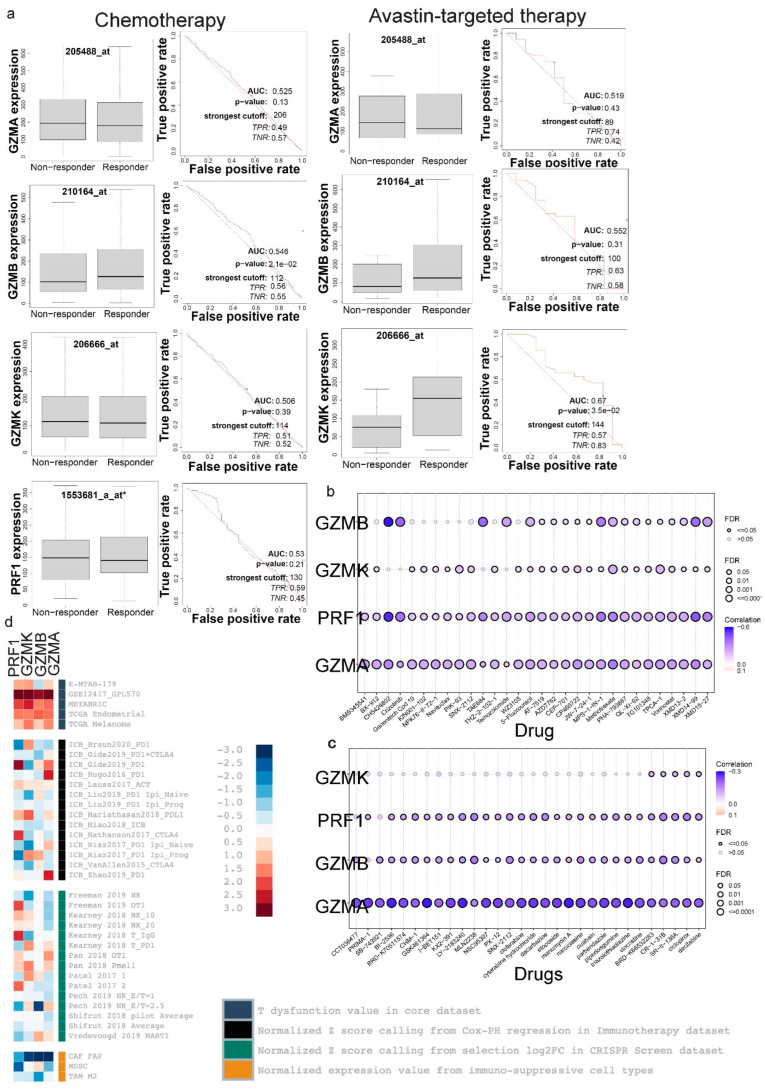
(**a**) Expression levels of *GZMA*, *GZMB*, *GZMK* and *PRF1* in responders and non-responders to chemotherapy and Avastin-targeted therapy to treat ovarian cancer. No results were available for *PRF1* expression in Avastin targeted ovarian cancer patients. (**b**,**c**) The bubble plot visualizes the correlations between the genes’ expression and drug sensitivity (GDSC and CTRP). Blue bubbles indicate negative correlations, while red bubbles indicate positive correlations, with deeper colors signifying stronger correlations. The bubble size corresponds to the FDR significance, and a black border indicates an FDR ≤ 0.05. (**d**) The heatmap depicts the results of regulator prioritization clustering and the relationships between *GZMA*, *GZMB*, *GZMK* and *PRF1* expression levels and immunosuppression.

**Figure 8 biomedicines-13-00408-f008:**
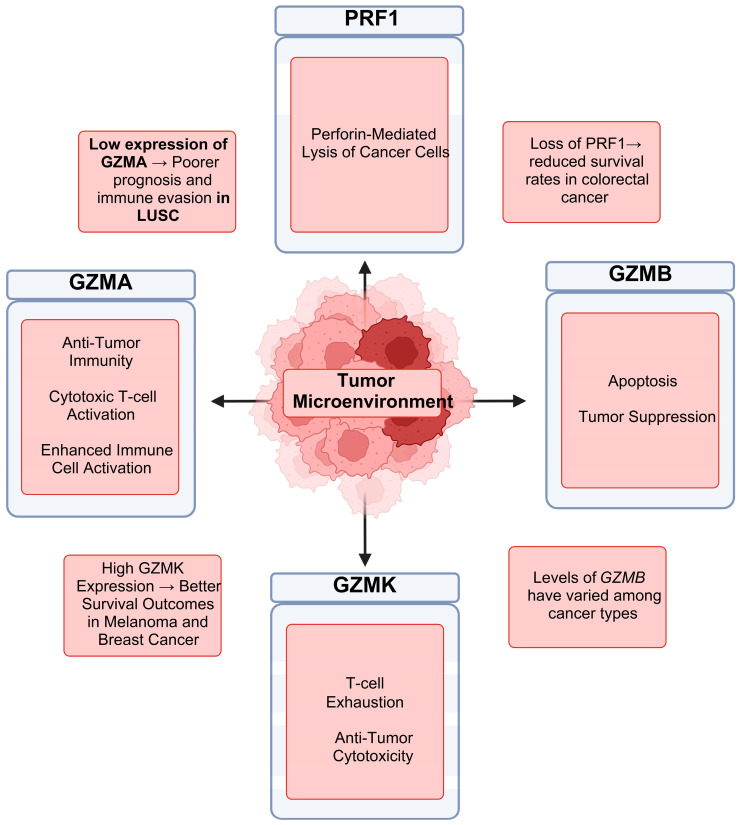
Roles of *GZMA, GZMB, GZMK* and *PRF1* in cancer development, growth and immune evasion. The major findings for each gene in the different cancer types are reported.

**Table 1 biomedicines-13-00408-t001:** The table depicts the name of the databases used, the number of cancers available on each platform, and the web location.

Database Name	Number of Cancers in the System	Database Link
Gene Set Cancer Analysis (GCSA)	33	https://guolab.wchscu.cn/GSCA//#/ (accessed on 10 May 2024)
Gene Expression Profiling Interactive Analysis 2 (GEPIA2)	33 (84 subtypes)	https://gepia2.cancer-pku.cn/ (accessed on 20 May 2024)
The Cancer Proteome Atlas (TCPA)	33	https://www.tcpaportal.org/tcpa/ (accessed on 10 May 2024)

**Table 2 biomedicines-13-00408-t002:** Relationship between GZMA and immune checkpoint blockade therapy response and survival. The table depicts the relationship between the expression of *GZMA* and the effectiveness of therapy in clinical trials involving immune checkpoint blockade from TIDE. In the Cohort column, “pre” and “post” refer to prior and post treatment, respectively. "Ipi_Prog" refers to patients who progressed after receiving Ipilimumab treatment, meaning their disease worsened despite therapy. "Ipi_Naive" refers to patients who have not previously received Ipilimumab treatment. These patients are either treatment-naïve in general or have received other therapies but not Ipilimumab.

Cohort	Cancer	Subtype	Cohort	Survival	CTL Cor	Risk	Adj. Risk	Count
Miao2018_ICB	Kidney	Clear		OS	0.956	0.177	−0.161	33
Braun2020_PD1	Kidney	Clear		OS	0.605	−1.082	−0.884	295
Riaz2017_PD1	Melanoma		Ipi_Naive	OS	0.94	−0.09	0.937	25
Liu2019_PD1	Melanoma		Ipi_Naive	OS	0.906	−0.371	−0.517	74
Riaz2017_PD1	Melanoma		Ipi_Prog	OS	0.97	−0.887	−0.537	26
Liu2019_PD1	Melanoma		Ipi_Prog	OS	0.927	−1.565	−0.177	47
Mariathasan2018_PDL1	Bladder	mUC		OS	0.931	−1.831	0.594	348
Nathanson2017_CTLA4	Melanoma		Post	OS	0.964	−1.819	0.782	15
Zhao2019_PD1	Glioblastoma		Post	OS	0.919	−1.69	−0.143	9
Nathanson2017_CTLA4	Melanoma		Pre	OS	0.97	0.443	−0.613	9
Zhao2019_PD1	Glioblastoma		Pre	OS	0.91	1.061	2.323	15
Lauss2017_ACT	Melanoma			OS	0.972	−0.43	0.236	25
Gide2019_PD1+CTLA4	Melanoma			OS	0.96	−0.854	0.305	32
Gide2019_PD1	Melanoma			OS	0.955	−2.121	1.687	41
VanAllen2015_CTLA4	Melanoma			OS	0.954	−2.526	−0.607	42
Hugo2016_PD1	Melanoma			OS	0.928	1.623	2.345	25
Miao2018_ICB	Kidney	Clear		PFS	0.956	0.849	−1.516	33
Braun2020_PD1	Kidney	Clear		PFS	0.605	−0.686	−0.606	295
Riaz2017_PD1	Melanoma		Ipi_Naive	PFS	0.94	0.8	1.857	25
Liu2019_PD1	Melanoma		Ipi_Naive	PFS	0.906	−0.613	0.213	74
Riaz2017_PD1	Melanoma		Ipi_Prog	PFS	0.97	−1.89	−1.388	26
Liu2019_PD1	Melanoma		Ipi_Prog	PFS	0.927	−2.122	0.391	47
Zhao2019_PD1	Glioblastoma		Post	PFS	0.919	−1.639	−0.325	9
Zhao2019_PD1	Glioblastoma		Pre	PFS	0.91	0.486	1.573	15
Lauss2017_ACT	Melanoma			PFS	0.972	−1.092	−1.303	25
Gide2019_PD1+CTLA4	Melanoma			PFS	0.96	−1.582	−0.464	32
Gide2019_PD1	Melanoma			PFS	0.955	−2.837	1.752	41
VanAllen2015_CTLA4	Melanoma			PFS	0.954	−1.927	−0.574	42
Prat2017_PD1	NSCLC-HNSC-Melanoma			PFS	0.762	−0.078	1.248	33

**Table 3 biomedicines-13-00408-t003:** Relationship between GZMB and immune checkpoint blockade therapy response and survival. The table depicts the relationship between the expression of *GZMB* and the effectiveness of therapy in clinical trials involving immune checkpoint blockade from TIDE. In the Cohort column, “pre” and “post” refer to prior and post treatment, respectively. "Ipi_Prog" refers to patients who progressed after receiving Ipilimumab treatment, meaning their disease worsened despite therapy. "Ipi_Naive" refers to patients who have not previously received Ipilimumab treatment. These patients are either treatment-naïve in general or have received other therapies but not Ipilimumab.

Cohort	Cancer	Subtype	Cohort	Survival	CTL Cor	Risk	Adj. Risk	Count
Riaz2017_PD1	Melanoma		Ipi_Prog	OS	0.948	−0.351	1.165	26
Gide2019_PD1	Melanoma			OS	0.9	−2.131	0.54	41
Liu2019_PD1	Melanoma		Ipi_Naive	OS	0.667	0.188	0.499	74
Lauss2017_ACT	Melanoma			OS	0.915	−0.253	0.434	25
Zhao2019_PD1	Glioblastoma		Post	OS	0.496	−0.148	0.366	9
VanAllen2015_CTLA4	Melanoma			OS	0.796	−1.755	0.319	42
Nathanson2017_CTLA4	Melanoma		Pre	OS	0.906	0.558	0.05	9
Riaz2017_PD1	Melanoma		Ipi_Naive	OS	0.964	−0.331	0.029	25
Zhao2019_PD1	Glioblastoma		Pre	OS	0.777	−0.029	−0.176	15
Nathanson2017_CTLA4	Melanoma		Post	OS	0.971	−1.917	−0.37	15
Liu2019_PD1	Melanoma		Ipi_Prog	OS	0.81	−1.431	−0.388	47
Mariathasan2018_PDL1	Bladder	mUC		OS	0.905	−2.175	−0.455	348
Hugo2016_PD1	Melanoma			OS	0.891	0.467	−0.623	25
Braun2020_PD1	Kidney	Clear		OS	0.708	−1.055	−0.679	295
Miao2018_ICB	Kidney	Clear		OS	0.813	−0.448	−0.696	33
Gide2019_PD1+CTLA4	Melanoma			OS	0.924	−1.319	−1.159	32
Riaz2017_PD1	Melanoma		Ipi_Prog	PFS	0.948	−0.742	2.228	26
Miao2018_ICB	Kidney	Clear		PFS	0.813	2.013	1.524	33
Zhao2019_PD1	Glioblastoma		Post	PFS	0.496	0.134	1.166	9
Lauss2017_ACT	Melanoma			PFS	0.915	−0.277	0.903	25
VanAllen2015_CTLA4	Melanoma			PFS	0.796	−1.209	0.518	42
Riaz2017_PD1	Melanoma		Ipi_Naive	PFS	0.964	0.482	0.501	25
Gide2019_PD1	Melanoma			PFS	0.9	−2.718	0.477	41
Zhao2019_PD1	Glioblastoma		Pre	PFS	0.777	0.145	0.397	15
Gide2019_PD1+CTLA4	Melanoma			PFS	0.924	−1.358	0.142	32
Prat2017_PD1	C-HNSC-Melanoma			PFS	0.867	−1.143	0.075	33
Liu2019_PD1	Melanoma		Ipi_Prog	PFS	0.81	−1.723	0.019	47
Liu2019_PD1	Melanoma		Ipi_Naive	PFS	0.667	−0.669	−0.14	74
Braun2020_PD1	Kidney	Clear		PFS	0.708	−0.729	−0.403	295

**Table 4 biomedicines-13-00408-t004:** Relationship between GZMK and immune checkpoint blockade therapy response and survival. The table depicts the relationship between the expression of *GZMK* and the effectiveness of therapy in clinical trials involving immune checkpoint blockade from TIDE. In the Cohort column, “pre” and “post” refer to prior and post treatment, respectively. "Ipi_Prog" refers to patients who progressed after receiving Ipilimumab treatment, meaning their disease worsened despite therapy. "Ipi_Naive" refers to patients who have not previously received Ipilimumab treatment. These patients are either treatment-naïve in general or have received other therapies but not Ipilimumab.

Cohort	Cancer	Subtype	Cohort	Survival	CTL Cor	Risk	Adj. Risk	Count
Mariathasan2018_PDL1	Bladder	mUC		OS	0.932	−1.421	1.818	348
Zhao2019_PD1	Glioblastoma		Post	OS	0.969	−1.421	1.668	9
Riaz2017_PD1	Melanoma		Ipi_Prog	OS	0.902	−0.263	1.442	26
Braun2020_PD1	Kidney	Clear		OS	0.536	−0.053	1.359	295
Lauss2017_ACT	Melanoma			OS	0.963	−0.38	0.437	25
Hugo2016_PD1	Melanoma			OS	0.916	0.865	−0.075	25
Liu2019_PD1	Melanoma		Ipi_Prog	OS	0.919	−1.469	−0.122	47
Zhao2019_PD1	Glioblastoma		Pre	OS	0.8	0.006	−0.128	15
Gide2019_PD1+CTLA4	Melanoma			OS	0.932	−1.03	−0.363	32
Miao2018_ICB	Kidney	Clear		OS	0.932	−0.28	−0.724	33
Nathanson2017_CTLA4	Melanoma		Post	OS	0.985	−1.954	−0.754	15
Nathanson2017_CTLA4	Melanoma		Pre	OS	0.973	0.343	−0.933	9
VanAllen2015_CTLA4	Melanoma			OS	0.964	−2.688	−1.427	42
Gide2019_PD1	Melanoma			OS	0.954	−2.672	−1.501	41
Liu2019_PD1	Melanoma		Ipi_Naive	OS	0.912	−0.754	−1.527	74
Riaz2017_PD1	Melanoma		Ipi_Naive	OS	0.941	−0.783	−1.829	25
Zhao2019_PD1	Glioblastoma		Post	PFS	0.969	−1.295	1.153	9
Braun2020_PD1	Kidney	Clear		PFS	0.536	0.451	0.972	295
Lauss2017_ACT	Melanoma			PFS	0.963	−0.561	0.822	25
Zhao2019_PD1	Glioblastoma	Pre		PFS	0.8	0.398	0.784	15
Miao2018_ICB	Kidney	Clear		PFS	0.932	1.465	0.395	33
Prat2017_PD1	C-HNSC-Melanoma			PFS	0.785	−1.025	0.192	33
Liu2019_PD1	Melanoma		Ipi_Prog	PFS	0.919	−2.088	−0.099	47
Gide2019_PD1+CTLA4	Melanoma			PFS	0.932	−1.473	−0.187	32
Liu2019_PD1	Melanoma		Ipi_Naive	PFS	0.912	−0.877	−0.514	74
Riaz2017_PD1	Melanoma		Ipi_Prog	PFS	0.902	−1.682	−0.919	26
VanAllen2015_CTLA4	Melanoma			PFS	0.964	−2.125	−1.313	42
Gide2019_PD1	Melanoma			PFS	0.954	−3.523	−1.671	41
Riaz2017_PD1	Melanoma		Ipi_Naive	PFS	0.941	−0.192	−2.283	25

**Table 5 biomedicines-13-00408-t005:** Relationship between PRF1 and immune checkpoint blockade therapy response and survival. The table depicts the relationship between the expression of *PRF1* and the effectiveness of therapy in clinical trials involving immune checkpoint blockade from TIDE. In the Cohort column, “pre” and “post” refer to prior and post treatment, respectively. "Ipi_Prog" refers to patients who progressed after receiving Ipilimumab treatment, meaning their disease worsened despite therapy. "Ipi_Naive" refers to patients who have not previously received Ipilimumab treatment. These patients are either treatment-naïve in general or have received other therapies but not Ipilimumab.

Cohort	Cancer	Subtype	Cohort	Survival	CTL Cor	Risk	Adj. Risk	Count
Lauss2017_ACT	Melanoma			OS	0.961	−0.281	0.784	25
Lauss2017_ACT	Melanoma			PFS	0.961	−0.762	−0.054	25
Riaz2017_PD1	Melanoma		Ipi_Prog	OS	0.961	−1.301	−2.047	26
Riaz2017_PD1	Melanoma		Ipi_Prog	PFS	0.961	−2.096	−2.054	26
Nathanson2017_CTLA4	Melanoma		Post	OS	0.96	−2.104	−1.274	15
Hugo2016_PD1	Melanoma			OS	0.94	0.793	−0.265	25
VanAllen2015_CTLA4	Melanoma			PFS	0.913	−1.673	0.005	42
VanAllen2015_CTLA4	Melanoma			OS	0.913	−2.462	−0.533	42
Miao2018_ICB	Kidney	Clear		OS	0.892	0.055	−0.27	33
Miao2018_ICB	Kidney	Clear		PFS	0.892	1.003	−0.473	33
Riaz2017_PD1	Melanoma		Ipi_Naive	OS	0.887	−0.723	−0.957	25
Riaz2017_PD1	Melanoma		Ipi_Naive	PFS	0.887	−0.057	−1.093	25
Gide2019_PD1+CTLA4	Melanoma			PFS	0.885	−1.258	0.136	32
Gide2019_PD1+CTLA4	Melanoma			OS	0.885	−1.025	−0.394	32
Liu2019_PD1	Melanoma		Ipi_Prog	PFS	0.86	−1.856	0.374	47
Liu2019_PD1	Melanoma		Ipi_Prog	OS	0.86	−1.536	−0.343	47
Mariathasan2018_PDL1	Bladder	mUC		OS	0.851	−1.221	1.084	348
Gide2019_PD1	Melanoma			OS	0.849	−1.163	2.802	41
Gide2019_PD1	Melanoma			PFS	0.849	−1.931	2.338	41
Liu2019_PD1	Melanoma		Ipi_Naive	OS	0.847	−0.411	−0.466	74
Liu2019_PD1	Melanoma		Ipi_Naive	PFS	0.847	−1.101	−0.85	74
Zhao2019_PD1	Glioblastoma		Post	OS	0.841	−1.8	−1.36	9
Zhao2019_PD1	Glioblastoma		Post	PFS	0.841	−1.889	−1.602	9
Zhao2019_PD1	Glioblastoma		Pre	OS	0.828	0.221	0.236	15
Zhao2019_PD1	Glioblastoma		Pre	PFS	0.828	−0.419	−0.567	15
Nathanson2017_CTLA4	Melanoma		Pre	OS	0.811	1.861	2.164	9
Prat2017_PD1	LC-HNSC-Melanoma			PFS	0.529	−1.184	−0.392	33
Braun2020_PD1	Kidney	Clear		PFS	0.118	−1.315	−1.251	295
Braun2020_PD1	Kidney	Clear		OS	0.118	−2.068	−1.924	295

## Data Availability

Data are contained within the article and Appendix A.

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
