# Peer review of "Comprehensive Analysis of Granzymes and Perforin Family Genes in Multiple Cancers"

_biomedicines, 2025, doi:10.3390/biomedicines13020408_

Round 1

Reviewer 1 Report

Comments and Suggestions for Authors

I am pleased to review the article titled as Comprehensive Analysis of Granzymes and Perforin Family Genes in Pan-Cancer”.

The authors have reported a comprehensive genomic analysis across various cancer types for the granzymes and perforin gene family, explored the differential expression, mutation profiles, and methylation patterns of these genes, focusing on their impact on tumor cell death and immune response modulation, providing insights into their potential as therapeutic targets.

It’s an interesting article and I would like the authors to address some of my comments prior to my endorsement of their manuscript for publication.

Following are my comments:

# Which approach was used for the Data normalization?

# Please provide link to the R package Survival in the material method section

# Please provide figure 1 in high resolution. So hard to read or interpret.

# It would be better to add at least one line explanation or summary of each table instead of mentioning them together as table 1-4.

# The conclusion section should focus on the findings of the article. I would recommend rewriting it completely.

Best wishes…

Author Response

# Which approach was used for the Data normalization?

Author response: We thank the reviewer for their question. Data normalization and batch correction were performed using RSEM (RNA-Seq by Expectation Maximization) for mRNA expression, ensuring comparability across samples and batches. Log2 transformation with TPM (Transcripts Per Million) normalization was employed for expression analyses, primarily utilizing data sourced from the GEPIA2 (Gene Expression Profiling Interactive Analysis 2) platform, which integrates TCGA (The Cancer Genome Atlas) and GTEx (Genotype-Tissue Expression) datasets. Pathway activity scores (PAS) were normalized using median centering and standard deviation scaling. We have now added this information to the methods section for clarification (lines 81-89 & 123-124).

# Please provide link to the R package Survival in the material method section

Author response: We thank the reviewer for this suggestion. We have now included the link to the Survival  R package in the material method section (line 169).

# Please provide figure 1 in high resolution. So hard to read or interpret.

Author response: We thank the reviewer for this feedback, we have now re-generated Figure 1 with the highest resolution, as requested.

# It would be better to add at least one line explanation or summary of each table instead of mentioning them together as table 1-4.

Author response: We thank the reviewer for their suggestion. We have revised the manuscript to include a one-line explanation or summary for each Table, providing a clear context for the data presented. We also have separated the Tables as 1a,b,c, and d, as requested by reviewer#2 (second comment).

# The conclusion section should focus on the findings of the article. I would recommend rewriting it completely.

Author response: We thank the reviewer for their feedback. We have completely re-written the conclusion to include the major findings of the article, its importance, and future steps (lines 895-921).

Best wishes…

Reviewer 2 Report

Comments and Suggestions for Authors

Comments to authors:

The manuscript of an article, which was written by Dr. Manvita Mareboina et al, is interesting, suggesting a possibility to apply in-silico gene expression pattern analysis on diagnosis of human diseases. Authors examined Granzyme/Perforin-encoding gene expression by in-silico analysis or GSCA platform to find that their expression is high in kidney and breast cancer. Authors further performed pathway analysis, immune filtration, SNV, CNV, and survival analyses to finally propose a possible diagram that the data mining in-silico system could be applied on personalized therapeutics or diagnosis of cancer utilizing gene expression/genomic DNA data. This article may contribute to quick and accurate clinical diagnosis of cancer in accordance with the progress in AI systems. However, I would suggest authors edit text and Figures to improve the manuscript suitable for publication as a scientific article.

Recommendation: Minor revision

General comments

1.       Figures had better be amended before publication. For all Figures, Titles on the top of panels and graphs should be eliminated but essential explanations remained for correct interpretation are needed. Such information can be usually shown on both vertical/horizontal axis, and Figure legend.

2.       Information of the personal data is not indicated. For example, the age and sex of the participants and malignancy of each cancer should be essentially required for interpretation of the data.

3.       The discussion is too long. Authors had better try to shorten that within one page and a half or so. Avoid redundant expressions but emphasize essential findings. Readers will wonder the reasons why expression of the GZMA/B/K and PRF1 genes are high in specific cancers. Therefore, authors had better add such discussion to develop better in-silico data-based diagnosis of cancer.

4.       Additionally, if authors successfully summarized discussion section, a schematic illustration that shows the importance of the GZMA/B/K and PRF1 on cancer generation/growth/malignancy can be added. Moreover, to suggest advantages of the diagnose cancer by in-silico data analyses, authors can present a diagram or a flow chart to be applied on the clinical purposes.

Specific comments

1.       Figures 1c and 2b: These Figures can be converted into Tables indicating averages with SD values and results of a statistical analysis.

2.       Tables should be renumbered as Table 1a, 1b, 1c, and 1d, instead of Table 1, 2, 3, and 4. Because they are showing the same data sets. However, the minimal but essential explanation of for survival, CTL Cor, Risk, Risk adj, and Count should be provided. It is described that Tables indicate the clinical trial data (P20, L707-709). If so, authors had better describe details for readers to understand the Tables correctly.

Minor comments

P3, L109, L117; P4, L189; P5, L221: Please check the font-size removing underlines.

P23, L721: [8,38, 39, 40, 41,42, 43]; [8, 38-43]

P2, L57: To avoid misunderstandings of DNA and protein, gene names should be typed in italic. Not only this part but also all through the text. Moreover, Gene IDs or link site of the NCBI data base would better be provided.

Author Response

Comments to authors:

The manuscript of an article, which was written by Dr. Manvita Mareboina et al, is interesting, suggesting a possibility to apply in-silico gene expression pattern analysis on diagnosis of human diseases. Authors examined Granzyme/Perforin-encoding gene expression by in-silico analysis or GSCA platform to find that their expression is high in kidney and breast cancer. Authors further performed pathway analysis, immune filtration, SNV, CNV, and survival analyses to finally propose a possible diagram that the data mining in-silico system could be applied on personalized therapeutics or diagnosis of cancer utilizing gene expression/genomic DNA data. This article may contribute to quick and accurate clinical diagnosis of cancer in accordance with the progress in AI systems. However, I would suggest authors edit text and Figures to improve the manuscript suitable for publication as a scientific article.

Recommendation: Minor revision

General comments

  1. Figures had better be amended before publication. For all Figures, Titles on the top of panels and graphs should be eliminated but essential explanations remained for correct interpretation are needed. Such information can be usually shown on both vertical/horizontal axis, and Figure legend.

Author response: Thank you for your valuable feedback. We have amended the figures to remove all the titles that did not need to be there and could be explained by the figure legends as well as figure axis. We have also removed additional redundant terms on the figures to ensure clarity.

  1. Information of the personal data is not indicated. For example, the age and sex of the participants and malignancy of each cancer should be essentially required for interpretation of the data.

Author response: We appreciate the reviewer’s attention to this important point regarding participant demographics. The data utilized in our study was obtained from The Cancer Genome Atlas (TCGA) through the GSCA database platform. While GSCA provides comprehensive genomic and clinical data including survival outcomes, disease staging, and molecular subtypes across 33 cancer types, detailed demographic information such as age and sex distributions is not directly available through the GSCA interface. The database focuses primarily on multi-dimensional genomic data (expression, SNV, CNV and methylation) and immunogenomic analyses of 24 immune cell types, along with clinical outcome measures. We acknowledge that demographic information would provide additional valuable context, but these specific parameters are not accessible within the current structure of the GSCA database system. However, these data are openly available through the TCGA platform.

  1. The discussion is too long. Authors had better try to shorten that within one page and a half or so. Avoid redundant expressions but emphasize essential findings. Readers will wonder the reasons why expression of the GZMA/B/Kand PRF1 genes are high in specific cancers. Therefore, authors had better add such discussion to develop better in-silico data-based diagnosis of cancer.

      Author response: Thank you for your valuable feedback regarding the discussion section. We have carefully revised and condensed the discussion to approximately one and a half pages, as requested, focusing on the most essential findings while eliminating redundant content. We have enhanced our discussion of the GZMA/B/K and PRF1 gene expression patterns in specific cancers, though we note that current literature in this area is still emerging. While very limited publications propose mechanisms for these expression patterns, they are mostly based on theoretical frameworks or preliminary findings that require further validation. We have carefully integrated these hypotheses where appropriate, while acknowledging the limitations of current evidence. The revised discussion maintains a focus on the implications for in-silico data-based cancer diagnostics, highlighting how our findings contribute to this growing field.

  1. Additionally, if authors successfully summarized discussion section, a schematic illustration that shows the importance of the GZMA/B/K and PRF1 on cancer generation/growth/malignancy can be added. Moreover, to suggest advantages of the diagnose cancer by in-silicodata analyses, authors can present a diagram or a flow chart to be applied on the clinical purposes.

Author response: Thank you for your valuable feedback. We have now addressed your suggestions by summarizing the discussion section more effectively and incorporating a new schematic illustration (new Figure 5) that highlights the roles of GZMA, GZMB, GZMK and PRF1 in cancer generation, growth, malignancy, and immune modulation. This figure visually emphasizes the importance of these immune-related genes in cancer pathogenesis and their potential clinical implications.

Specific comments

  1. Figures 1c and 2b: These Figures can be converted into Tables indicating averages with SD values and results of a statistical analysis.

Author response: Thank you for this thoughtful suggestion regarding the presentation of data in Figures 1c and 2b. The visualizations presented in our manuscript reflect the standardized graphical outputs provided by the GSCA database interface, which generates these specific graphic formats to display the analytical results. While we appreciate the value of tabular presentation with statistical measures as suggested, we believe that the current graphical representations, though not in tabular format, effectively communicate the statistical relationships and patterns in our data. These visualizations are designed to provide clear and interpretable demonstrations of the results, consistent with how the GSCA database presents such analyses. Additionally, we hoped to keep consistency in the way we presented our data in graphical formats across all of our analyses.

  1. Tables should be renumbered as Table 1a, 1b, 1c, and 1d, instead of Table 1, 2, 3, and 4. Because they are showing the same data sets. However, the minimal but essential explanation of for survival, CTL Cor, Risk, Risk adj, and Count should be provided. It is described that Tables indicate the clinical trial data (P20, L707-709). If so, authors had better describe details for readers to understand the Tables correctly.

Author response: Thank you for your thoughtful feedback. We have revised the manuscript to include minimal but essential explanations for survival, CTL correlation, risk, risk adjustment, and count as they pertain to the tables. Specifically, we have expanded on the descriptions in the main text to clarify the clinical trial data presented in Tables 1a-d and how they relate to immune checkpoint blockade outcomes. These Tables now provide detailed insights into the correlation of gene expression with overall survival (OS) and progression-free survival (PFS) across various cancers, as well as risk adjustments and immune indicators.

Minor comments

P3, L109, L117; P4, L189; P5, L221: Please check the font-size removing underlines.

Author response: We thank the reviewer for their feedback. All font-size removed underlines have been checked.

P23, L721: [8,38, 39, 40, 41,42, 43]; [8, 38-43]

Author response: We thank the reviewer for their feedback. The citation format has been updated.

P2, L57: To avoid misunderstandings of DNA and protein, gene names should be typed in italic. Not only this part but also all through the text. Moreover, Gene IDs or link site of the NCBI data base would better be provided.

Author response: We thank the reviewer for their feedback. The manuscript has been updated to italicize gene names. Gene names have been updated to be italicized throughout the manuscript. The link site of the NCBI database has been added to lines 78-79 (P2).